# Early macrophage response to obesity encompasses Interferon Regulatory Factor 5 regulated mitochondrial architecture remodelling

L. Orliaguet [1,2], T. Ejlalmanesh[1,2], A. Humbert [3], R. Ballaire[1,2], M. Diedisheim[1,2,4], J. B. Julla[1,2,5], D. Chokr[1,2], J. Cuenco[1,2], J. Michieletto[6], J. Charbit[7], D. Lindén [8], J. Boucher[8], C. Potier[1,2], A. Hamimi[2], S. Lemoine [9], C. Blugeon[9], P. Legoix[10], S. Lameiras[10], L. G. Baudrin[10], S. Baulande [10], A. Soprani[1,2,11], F. A. Castelli[6], F. Fenaille [6], J. P. Riveline[1,2,5], E. Dalmas[1,2], J. Rieusset[3], J. F. Gautier[1,2,5], N. Venteclef [1,2] ✉ & F. Alzaid [1,2,12] ✉

Adipose tissue macrophages (ATM) adapt to changes in their energetic microenvironment. Caloric excess, in a range from transient to diet-induced obesity, could result in the transition of ATMs from highly oxidative and protective to highly inflammatory and metabolically deleterious. Here, we demonstrate that Interferon Regulatory Factor 5 (IRF5) is a key regulator of macrophage oxidative capacity in response to caloric excess. ATMs from mice with genetic-deficiency of *Irf5* are characterised by increased oxidative respiration and mitochondrial membrane potential. Transient inhibition of IRF5 activity leads to a similar respiratory phenotype as genomic deletion, and is reversible by reconstitution of IRF5 expression. We find that the highly oxidative nature of *Irf5*-deficient macrophages results from transcriptional de-repression of the mitochondrial matrix component Growth Hormone Inducible Transmembrane Protein (GHITM) gene. The *Irf5*-deficiency-associated high oxygen consumption could be alleviated by experimental suppression of *Ghitm* expression. ATMs and monocytes from patients with obesity or with type-2 diabetes retain the reciprocal regulatory relationship between *Irf5* and *Ghitm*. Thus, our study provides insights into the mechanism of how the inflammatory transcription factor IRF5 controls physiological adaptation to diet-induced obesity via regulating mitochondrial architecture in macrophages.

Macrophage metabolism is a powerful mitigating or optimising factor influencing function[1]. Generally speaking, pro-inflammatory polarisation relies on glycolysis, mediated by hypoxia-inducible factor (HIF) $-1\alpha$[2], with specific interruptions of the tricarboxylic acid (TCA) cycle[3]. Conversely, anti-inflammatory polarisation or regulatory function is supported by mitochondria and oxidative respiration[4]. Microenvironmental niches also impose energetic specificities on macrophages[5,6]. For example, adipose tissue macrophages (ATM) can range from being

metabolically quiescent to overall hypermetabolic[5,7,8]. In diet-induced obesity (DIO), ATMs are exposed to the same dysmetabolism as all peripheral tissues, that is glucolipotoxicity, providing a systemic abundance of metabolic substrates. It is in this context that ATMs are hypermetabolic, with highly glycolytic and highly oxidative fluxes[7–9]. ATMs are, however, a heterogenous population of cells that exhibit a range of beneficial and detrimental phenotypes over the course of obesity[10,11]. Lipid-associated macrophages (LAM) have been characterised, they are highly oxidative and have a high capacity to clear lipids and dying adipocytes[12]. Also highly responsive to lipids are the phenotypically similar MARCO+ lipid-buffering ATMs and metabolically activated macrophages (MMe)[13,14]. Expansion of these ATM populations and their functional contribution to maintaining tissue homeostasis or to metabolic decline varies with the duration of caloric excess[13]. Such reports indicate that macrophages are reactive beyond their inflammatory roles, and their successful adaptation early in the course of DIO, may be sufficient to mitigate systemic metabolic decline.

The rise of lipid-buffering ATMs represents initial adaptation in a physiological attempt to maintain homeostasis[10]. Over prolonged caloric excess ATMs become predominantly inflammatory and contribute to insulin resistance in DIO[15,16]. The early adaptive step is marked by high oxidative capacity[14], and the later predominance of inflammatory ATMs marks their rise as critical actors in the development of insulin resistance and type-2 diabetes (T2D)[13].

The Interferon Regulatory Factor (IRF)−5 is a key molecular switch mediating M1-like polarisation of ATMs[17]. ATM expression of IRF5 is increased in DIO, promoting pro-inflammatory polarisation and repressing TGFβ-signalling. This favours maladaptive white adipose tissue (WAT) expansion and insulin resistance. IRF5 is physiologically required to respond to bacterial and viral stimuli[18,19], evidence also implicates deregulated expression in conditions of chronic inflammation (e.g. auto-immune, metabolic diseases)[20,21]. Gain-of-function *Irf5* variants associated with the auto-immune disease have more recently been found to promote macrophage glycolytic programming[22].

Here, we reveal a non-canonical function for IRF5 in orienting macrophage energetic adaptation to caloric excess. IRF5 transcriptionally represses the gene encoding *Growth hormone inducible transmembrane protein* (*Ghitm*), a key mitochondrial component required for oxidative respiration. Through this interaction, IRF5 contributes to failure in maintaining normal mitochondrial cristae structures that support effective oxidative respiration. GHITM repression and failure to maintain cristae structure restrain ATM oxidative capacity in DIO. The IRF5-GHITM regulatory axis extends from short- to long-term high-fat feeding and to monocytes and ATMs in patients with obesity and T2D.

## Results

### IRF5 is associated with ATM metabolic adaptation upon short-term high-fat diet

We started by analysing the ATM transcriptome from mice with a myeloid-deficiency of *Irf5* (IRF5-KO) or wild-type (WT) mice on 4 and 12 weeks of a high-fat diet (HFD). On a 12-week, long-term HFD (LT-HFD), differentially expressed genes were associated with inflammatory response and tissue remodelling (Fig. 1a, S1A). This confirms previously reported phenotypic features[17]. On a 4-week HFD, qualified short term (ST-HFD), differentially expressed genes enriched several GO terms for metabolic process. Interestingly, terms relating to immune function (humoral immune response, phagocytosis/recognition) were under-represented (Fig. 1b, S1B). These results indicate that IRF5 may influence ATM metabolic adaptation, in particular, in response to short-term caloric excess.

To associate ATM metabolism with IRF5, we evaluated ATM metabolic adaptation and IRF5 expression upon ST- and LT-HFD in C57BL/6 J mice. Mice on ST-HFD and LT-HFD gained weight, increasing

WAT mass and losing glycaemic homeostasis over time (Fig. S1C–E). IRF5 expression also increased in epididymal fat pads (EpiWAT) on ST-HFD and LT-HFD (Fig. 1c). We characterised ATM metabolic adaptation using the fluorescent lipid dye BODIPY and the JC-1 dye, a sensor for mitochondrial mass (Mt Mass) and membrane potential (mΔΨ)[23]. On ST- and LT-HFD, ATMs have a higher lipid content and Mt Mass but decreased mΔΨ and mΔΨ-to-mass ratio, relative to mice on a normal chow diet (NCD) (Fig. 1d, S2A). Interestingly, effects on Mt Mass and mΔΨ upon ST-HFD are similar in magnitude to LT-HFD. These data are consistent with previous reports that ATMs become hypermetabolic in DIO[7,8]; however, metabolic adaptation occurs within short-term caloric excess. This was confirmed to contribute to cellular respiration by extracellular flux analyses on F4/80+ ATMs (Fig. 1e, S2B).

On ST-HFD, correlations revealed that ATM Mt Mass was positively associated with IRF5 expression and ΔΨ-to-mass ratio was negatively associated (Fig. 1f), ATM lipid content was not associated (Fig. S2C). ATM numbers were positively correlated to IRF5 expression, this was observed by FACS and by qPCR analysis of F4/80 and IRF5 expression in EpiWAT (Fig. 1g, S2D). A UMAP showed that IRF5 was highly expressed in cells that also highly express F4/80 (F4/80Hi; Fig. S2E), a population reported to be monocyte-derived[24]. We quantified IRF5 expression in F4/80Lo and F4/80Hi ATMs and found IRF5 to be upregulated in F4/80Hi ATMs on ST-HFD (Fig. 1h; S2F). F4/80Hi ATMs had markedly increased Mt Mass and decreased mΔΨ (Fig. 1i). These results suggest that IRF5 plays a role in ATM mitochondrial adaptation, in particular in F4/80Hi ATMs.

### IRF5 deficiency alters ATM oxidative respiration in response to a short-term high-fat diet

We applied the same model of ST-HFD to IRF5-KO and WT mice. Weight gain and EpiWAT weight were similar between genotypes (Fig. S3A). ATMs from IRF5-KO mice had increased mΔΨ and mΔΨ-to-mass ratio relative to WT mice, lipid content and Mt Mass were not altered (Fig. 1j; Fig. S3B). Analysis by tSNE confirmed JC1-red fluorescence, indicating mΔΨ, was highest in F4/80Hi ATMs and these cells had higher fluorescence in IRF5-KO (Fig. 1k). Under basal conditions, on NCD, IRF5-KO did not affect ATM metabolic phenotype (Fig. S3C), and upon LT-HFD, only a trend to increased intracellular lipid content persisted (Fig. S3D).

To link cytometric analyses to functional respiration, we analysed extracellular flux from magnetically sorted F4/80+ ATMs of IRF5-KO and WT mice under NCD and following ST- and LT-HFD. ATMs from IRF5-KO mice had a higher oxygen consumption rate (OCR) following ST-HFD, but not on NCD nor LT-HFD (Fig. 2a). F4/80− cells were unaffected by IRF5 deficiency (Fig. S4A). However, higher OCR in IRF5-KO ATMs remains apparent when whole SVF is analysed under conditions testing mitochondrial, or glycolytic, respiration (Fig. 2b, c). OCR reflects a number of oxygen-consuming processes, a major contributor to which is fatty acid oxidation (FAO)[25]. To evaluate the contribution of FAO to the IRF5-KO respiratory phenotype, we carried out a palmitate oxidation test on SVF, with or without etomoxir, an inhibitor of carnitine palmitoyl transferase (CPT)−1[26]. OCR was higher in palmitate-loaded SVF from IRF5-KO mice upon ST-HFD. This was normalised to WT levels in the presence of etomoxir (Fig. 2d, S4B), indicating that FAO contributes to higher OCR in cells from IRF5-KO mice.

### ATM adaptation in IRF5 deficiency alters adipose tissue phenotypic response to a short-term high-fat diet

As a consequence of ATM phenotype, analysing EpiWAT sections revealed that average adipocyte diameter and frequency of large (>100 um) adipocytes were higher in EpiWAT from IRF5-KO mice (Fig. 2e, S4C). The number of crown-like structures (CLS) was higher, and the number of MAC2+ cells had an increasing trend (Fig. 2f, S4D). Despite CLS accumulation in IRF5-KO, we found no difference in the expression of inflammatory markers in EpiWAT, (e.g. IL6, TNF, some

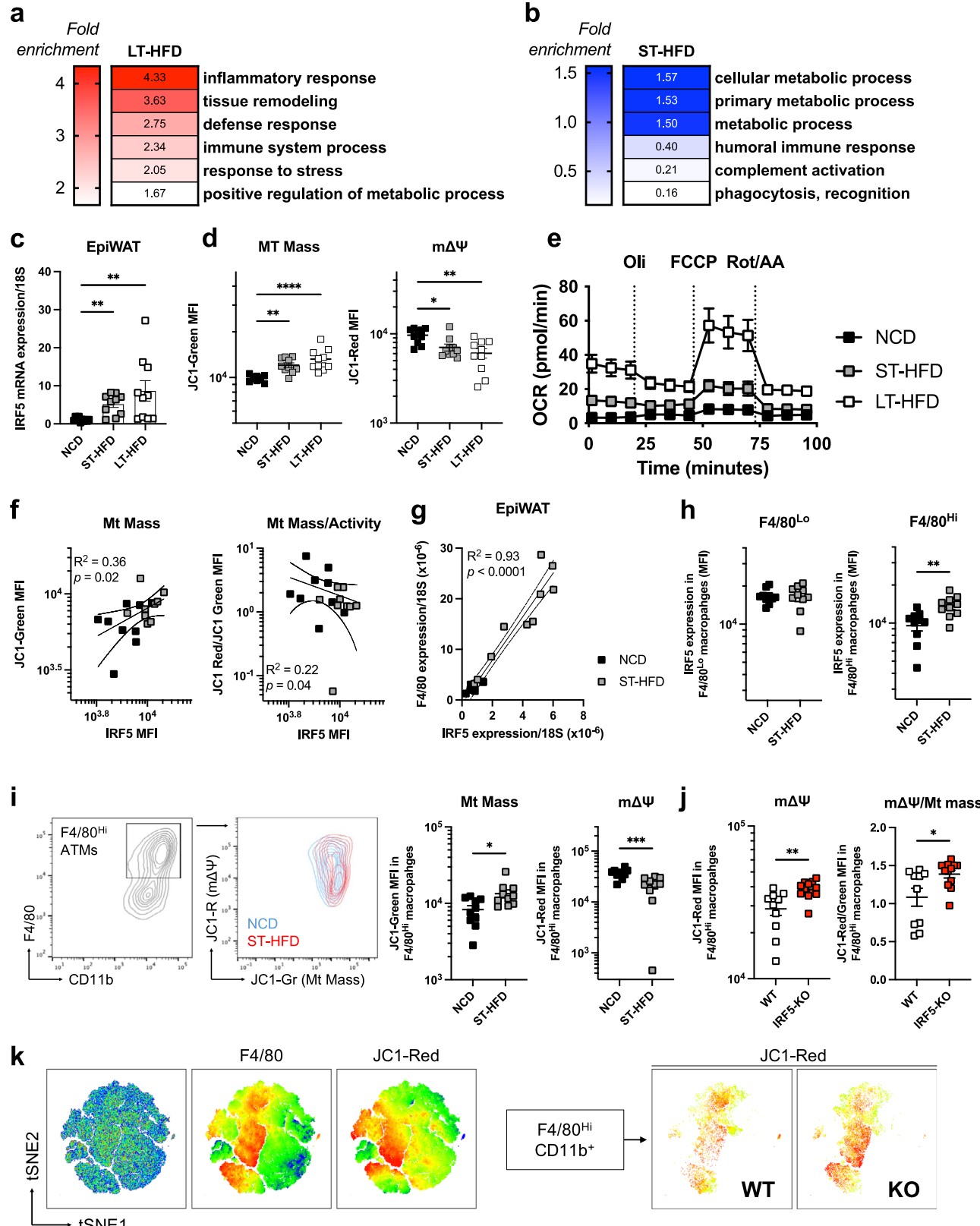

markers were not reliably detectable). Cytokine and adipokine levels in circulation were also similar between genotypes (Fig. S4E–G). These phenotypic tissue features were concurrent to a functional increase in glucose uptake capacity in fat pads from IRF5-KO mice (Fig. 2g), and this occurs on a background of similar glycaemia and insulin levels to WT mice (Fig. S4H). Increased glucose uptake might reflect improved glucose homeostasis and insulin sensitivity at the tissue level and could

also explain increased adipocyte size in IRF5-KO mice. The EpiWAT phenotype of IRF5-KO mice upon ST-HFD presents similarities with the protective EpiWAT phenotype of IRF5-KO mice upon LT-HFD[17] (i.e. increased ATM content, improved insulin sensitivity). Importantly, the IRF5-linked respiratory phenotype of ATMs occurs transiently and orchestrates tissue level adaptation at a stage when systemic metabolism is not yet impacted (Fig. S4I). Such early adaptation,

**Fig. 1 | IRF5 expression is associated with ATM metabolic adaptation to short-term caloric excess. a, b** Wild-type (WT) mice and mice with myeloid-deficiency of IRF5 (IRF5-KO) were placed on (**a**) 12-week long-term (LT-) high-fat diet (HFD) or (**b**) 4-week short-term (ST-)HFD. F4/80+ adipose tissue macrophages (ATM) were sorted from epididymal fat pads (EpiWAT) for RNA-seq. Differentially expressed genes between genotypes were used for gene ontology analysis ($n = 4$ per genotype, Wald test $p$-value <0.05, Log$_2$FC > 1.0). Heatmaps represent fold-enrichment (Binomial test of gene list compared to reference genome, $p$ values in Fig. S1A, B). **c** Irf5 expression in EpiWAT from C57BL/6 J mice on normal chow diet (NCD), ST-HFD or LT-HFD ($n = 10$ per group; Kruskal–Wallis multiple comparisons, left-to-right **$p = 0,0063$; **$p = 0,0015$). **d** JC1-Green and JC1-Red fluorescence to assess mitochondrial mass (MT Mass) and membrane potential (mΔΨ) in ATMs of C57BL/6 J mice on NCD, ST-HFD or LT-HFD ($n = 10$ per group on NCD/ST-HFD; $n = 7$ on LT-HFD; one-way ANOVA, **$p = 0.0079$; ****$p = 0,000094$). **e** Oxygen consumption rate (OCR) from ATMs of C57BL/6 J mice on NCD, ST-HFD or LT-HFD. Cells treated with Oligomycin (Oli), carbonyl cyanide 4-(trifluoromethoxy)

phenylhydrazone (FCCP) and Rotenone/Antimycin A (Rot/AA) ($n = 4$ on NCD, $n = 9$ on ST-HFD, $n = 8$ on LT-HFD). **f** Correlation between IRF5 and JC1-Green MFI (Linear regression, Pearson correlation R$^2$ = 0.36; $p = 0.02$) and JC1-Red/JC1-Green MFI (R$^2$ = 0.22; $p = 0.04$) in ATMs of C57BL/6 J mice on NCD or ST-HFD ($n = 9$ per group). **g** Correlation between Irf5 and F4/80 expression in EpiWAT of C57BL/6 J mice on NCD or ST-HFD ($n = 10$ per group). **h** IRF5 MFI in F4/80$^{Lo}$ and F4/80$^{Hi}$ ATMs of C57BL/6 J mice on NCD or ST-HFD ($n = 10$ per group, two-tailed unpaired $t$-test, **$p = 0.004$). **i** Flow cytometry contour plots of JC1-Green and JC1-Red in F4/80$^{Hi}$ ATMs of C57BL/6 J on NCD or ST-HFD. MFI of JC1-Green and JC1-Red in F4/80$^{Hi}$ ATMs ($n = 10$ mice per group, two-tailed unpaired $t$-tests, *$p = 0.01$; ***$p = 0.0009$). **j** MFI of JC1-Red and JC1-Red/JC1-Green in F4/80$^{Hi}$ EpiWAT ATMs of WT and IRF5-KO mice on ST-HFD ($n = 10$ for WT and 12 for IRF5-KO, two-tailed unpaired $t$-tests, **$p = 0.0032$; *$p = 0.019$). **k** tSNE plot of F4/80 and JC1-Red MFI on the stromal vascular fraction of EpiWAT, tSNE plot of JC1-Red MFI on F4/80$^{Hi}$ populations from mice on ST-HFD. Data presented as mean ± SEM. Source Data file provided.

underpinned by ATM mitochondrial respiration, is a key event that precedes IRF5 deficiency's protective metabolic phenotype at the stage of systemic insulin resistance (LT-HFD).

## IRF5 repression of mitochondrial respiration is cell intrinsic, reversible and inducible in mature macrophages

To carry out mechanistic investigations, we moved to bone-marrow-derived macrophages (BMDM). BMDMs from IRF5-KO and WT mice were differentiated and treated for 24 h with bacterial lipopolysaccharides (LPS), a canonical stimulant of the IRF5 signalling pathway, or with palmitate to model lipotoxicity. Testing glycolysis, we found no genotype difference in extracellular acidification rates (ECAR) in control or treated cells (Fig. S5A). Glucose-stimulated OCR was increased in IRF5-KO BMDMs following treatment with LPS or palmitate (Fig. 3a, S5B). Under conditions testing mitochondrial respiration, OCR was increased in IRF5-KO BMDMs following LPS or palmitate treatment, with no difference in untreated cells (Fig. 3b, S5C). The IRF5-linked respiratory phenotype is cell intrinsic and mirrors what we observed in ATMs.

To evaluate whether the respiratory phenotype is the result of genetic deficiency or if it is inducible in mature macrophages, we applied an IRF5 inhibitory decoy peptide (IRF5-DP) to mature BMDMs from WT mice. IRF5-DP binds to IRF5, preventing its nuclear translocation[27]. LPS-induction of TNF is prevented by IRF5-DP, confirming that it blocks the transcriptional activity of IRF5 (Fig. S5D). When treated with palmitate, metabolic flux analyses showed that IRF5-DP increased OCR relative to the vehicle, replicating the effect of genetic deficiency (Fig. 3c). This result indicates a requirement for IRF5 nuclear translocation and rules out a differentiation effect of genetic deficiency. We next used adenoviral delivery to re-introduce IRF5 expression in BMDMs from IRF5-KO mice, IRF5 adenovirus (adIRF5) resulted in a 1.4-fold increase in IRF5 expression (Fig. S5F). Following palmitate treatment, OCR was decreased in cells treated with adIRF5, but not in cells treated with the control adenovirus (adGFP; Fig. 3d, S5G).

## IRF5 deficiency alters concentrations of TCA cycle metabolites and structural components of mitochondria in response to palmitate treatment

To understand how IRF5 affects mitochondrial function, we quantified TCA cycle metabolites in IRF5-KO, and WT BMDMs treated with palmitate or with LPS. A PCA score plot revealed a genotype-dependent difference in metabolite profile within 2 h of treatment with palmitate but not with LPS, and most differences were normalised by 24 h (Fig. 4a, S6A, B). This was confirmed by carrying out a PCA only on 2 h Palm-treated samples (Fig. 4b). Variable ranking revealed lactate was the biggest contributor to the IRF5-dependent response to palmitate, and it had a higher concentration in IRF5-KO BMDMs (Fig. 4c). Lactate is a glycolysis end-product destined for extracellular release[28] (Fig. 4d).

However, ECAR was lower in IRF5-KO BMDM under these test conditions, indicating lactate is released at a slower rate (Fig. 4e). Thus, lactate accumulation in IRF5-KO BMDM can be explained by its increased retention. Interestingly, intracellular lactate has recently been reported to be subject to oxidation in M2-like macrophages, potentially contributing to oxygen consumption[29]. Consequently, analysing mitochondrial respiration found increased OCR in IRF5-KO BMDM under these conditions.

We also applied electron microscopy to BMDMs under these same conditions to evaluate potential structural mechanisms. Mitochondrial density, form factor and aspect ratio were not altered between IRF5-KO and WT BMDM (Fig. 4g, S6C), suggesting no adaptation in mitochondrial dynamics as quantifiable by these parameters. However, we did find that mitochondrial cristae were denser and had a larger surface area in IRF5-KO BMDM (Fig. 4h). Well-developed cristae structures may provide a mechanism to increase the surface area for oxidative respiration, allowing IRF5-KO BMDM to maintain their hyperoxidative phenotype.

The above analyses highlight two potential mechanisms that can contribute to increased oxidative respiration in IRF5-deficient macrophages: (1) increase in oxidisable intracellular lactate, or (2) increase in mitochondrial respiratory surface area.

## IRF5 binds and regulates the expression of the mitochondrial matrix protein GHITM

To resolve a transcriptional mechanism, we carried out RNA-seq on ATMs (ST- and LT-HFD) and BMDM (0, 2 and 24 h stimulation with LPS or Palm) from IRF5-KO and WT mice. Coregulated clusters were defined based on genotype effect and on a trajectory over time (Fig. 5a, S7A). A number of terms relating to lipotoxicity and mitochondrial function were enriched across all conditions (e.g. response to cholesterol, mitochondrial translation; Fig. 5b). We also acquired ChIP-seq data that maps IRF5 binding in BMDM[30]. Of 526 bound genes, 77 (14.6%) enriched the metabolic process GO term (Fig. 5c), indicating a level of transcriptional control over metabolism.

To define a list of targets, we carried out differential expression analyses of RNA-seq data between genotypes, per condition and per timepoint (Fig. 5d). Palmitate treatment had the highest number of differentially expressed genes, followed by LPS and HFD; 34 targets (1%) were represented in all conditions. Intersection with ChIP-seq revealed 6 genes were differentially expressed and were also bound by IRF5 at, or upstream of, transcription start sites: *Atf5, Syce2, Abcg1, Lrrc27, Fnip2* and *Ghitm*. GHITM has an overt function in maintaining inner membrane cristae structures[31] (Fig. 5d), and its expression was negatively correlated with *Irf5* expression in ATMs from WT mice (Fig. 5e). These data indicate that *Ghitm* may be a mechanistic target of IRF5 that can influence mitochondrial respiration.

Single-cell sequencing data[12] from mice on HFD confirmed that *Irf5* and *Ghitm* are highly expressed in monocytes and ATMs, *Ghitm*

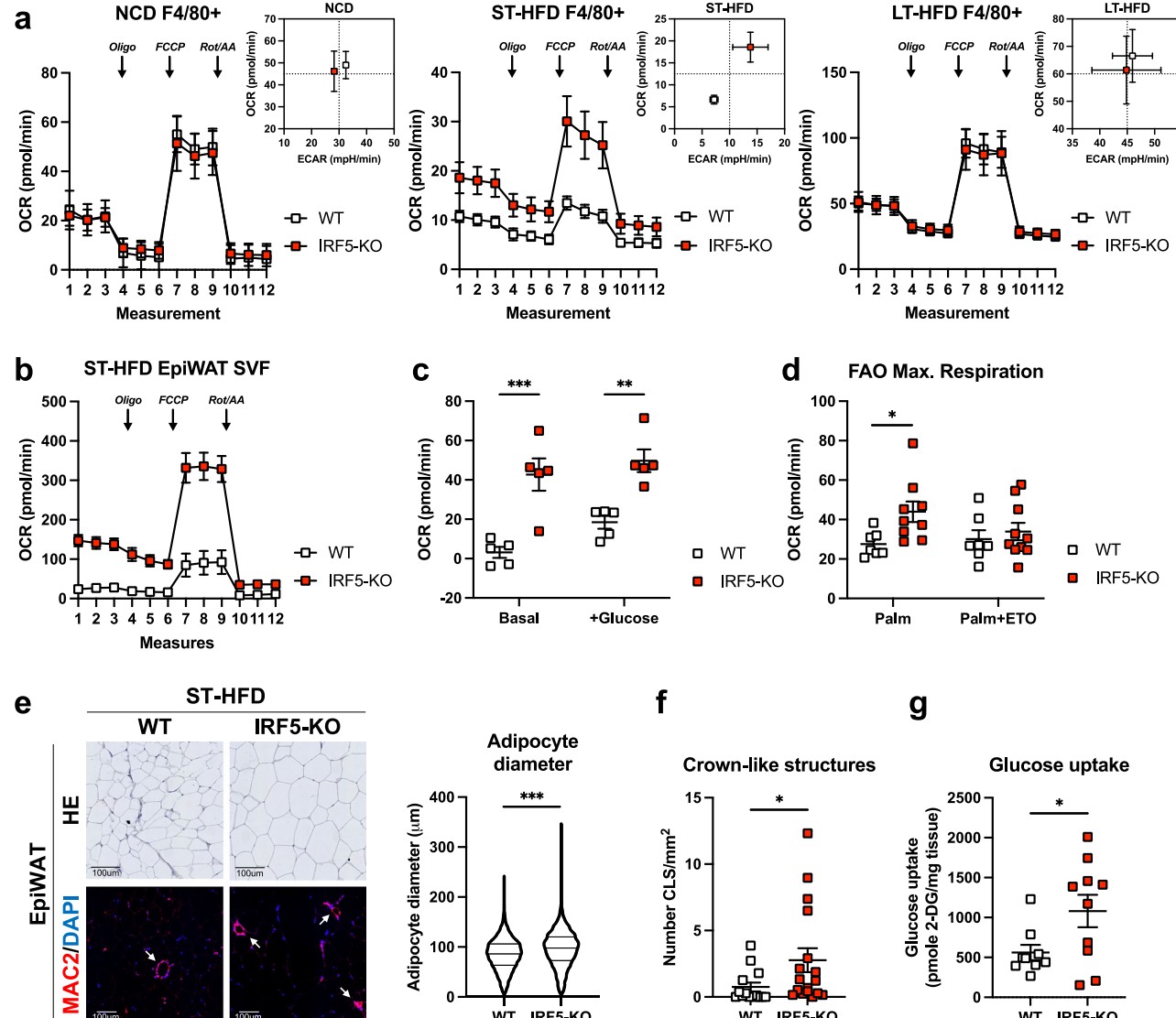

**Fig. 2 | Increased mitochondrial respiration in IRF5-deficient macrophages alters adipose tissue phenotype and function upon short-term high-fat diet.**
**a** Oxygen consumption rate (OCR) from extracellular flux analysis, following Oligomycin (Oli), carbonyl cyanide 4-(trifluoromethoxy) phenylhydrazone (FCCP) and Rotenone/Antimycin A (Rot/AA) treatments, from epididymal white adipose tissue (EpiWAT) magnetically sorted F4/80⁺ cells from WT and IRF5-KO mice on normal chow diet (NCD; left), short-term (ST-) high-fat diet (HFD; middle) and long-term (LT-)HFD (right). Energetic plot with Extracellular acidification rate (ECAR) and OCR from maximal respiration ($n = 4$ mice per genotype for NCD, $n = 6$ WT and 8 IRF5-KO mice for ST-HFD and $n = 5$ WT and 4 IRF5-KO mice for LT-HFD). **b** OCR from extracellular flux analysis, following Oligo, FCCP and Rot/AA treatments, from EpiWAT stromal vascular fraction (SVF) of WT and IRF5-KO mice on ST-HFD ($n = 5$ mice per genotype). **c** OCR from extracellular flux analysis under basal conditions and following addition of glucose, performed on the EpiWAT SVF of WT and IRF5-KO mice on ST-HFD ($n = 5$ mice per genotype, ***$p = 0.0002$

and **$p = 0.0018$ two-way ANOVA). **d** Maximal OCR from Fig. S4B of fatty acid oxidation (FAO) test on EpiWAT SVF from WT ($n = 7$) and IRF5-KO ($n = 9$) mice on ST-HFD (*$p = 0.035$, one-way ANOVA). Palm palmitate, ETO etomoxir.
**e** Representative images of hematoxylin and eosin (HE) staining for adipocyte size and MAC2/DAPI immunostaining to visualise crown-like structures (white arrows), on EpiWAT sections of WT and IRF5-KO mice on ST-HFD (scale bar = 100 um). Right: adipocyte size quantification on the HE staining (data pooled from biologically independent replicates, $n = 20$ WT and 21 IRF5-KO mice, two-tailed unpaired $t$-test, ***$p = 0.0001$). **f** Crown-like structure quantification on MAC2/DAPI immunostained EpiWAT sections of WT and IRF5-KO mice on ST-HFD ($n = 14$ WT and 17 IRF5-KO mice, two-tailed unpaired $t$-test Welch's correction, *$p = 0.0472$). **g** Glucose uptake assay performed on EpiWAT explants from WT and IRF5-KO mice on ST-HFD ($n = 9$ WT and 10 IRF5-KO mice, two-tailed unpaired $t$-test Welch's correction, $p = 0.038$). Data presented as mean ± SEM. Source Data file provided.

expression was higher than all other targets identified (Figs. 6a, b, S8A). *Irf5* and *Ghitm* expression were negatively correlated (R = −0.44; $p < 0.001$), supporting our own data (Fig. 6c, d). *Irf5* expression increased over time and remained negatively correlated to *Ghitm* expression in monocytes and ATMs (Fig. 6b, d). Data analysed from Jaitin et al.[12], Saliba et al.[30], together with our work, strongly suggest that the interaction between IRF5 and *Ghitm* influences macrophage respiratory phenotype.

We chose to pursue a GHITM-mediated mechanism as a contributor to increased oxygen consumption through the maintenance

of cristae structures in IRF5-deficient macrophages (Fig. 4h). As for the potential contribution of lactate oxidation, none of the identified targets had a described function in lactate metabolism, and thus IRF5-dependent remodelling of the TCA cycle could be an area for future investigation beyond the scope of current work.

## GHITM knockdown reverses hyperoxidative phenotype of IRF5-deficient macrophages
With guide RNAs (gRNA) targeting *Ghitm* (gGHITM), we transduced BMDM expressing the Clustered Regularly Interspaced Short

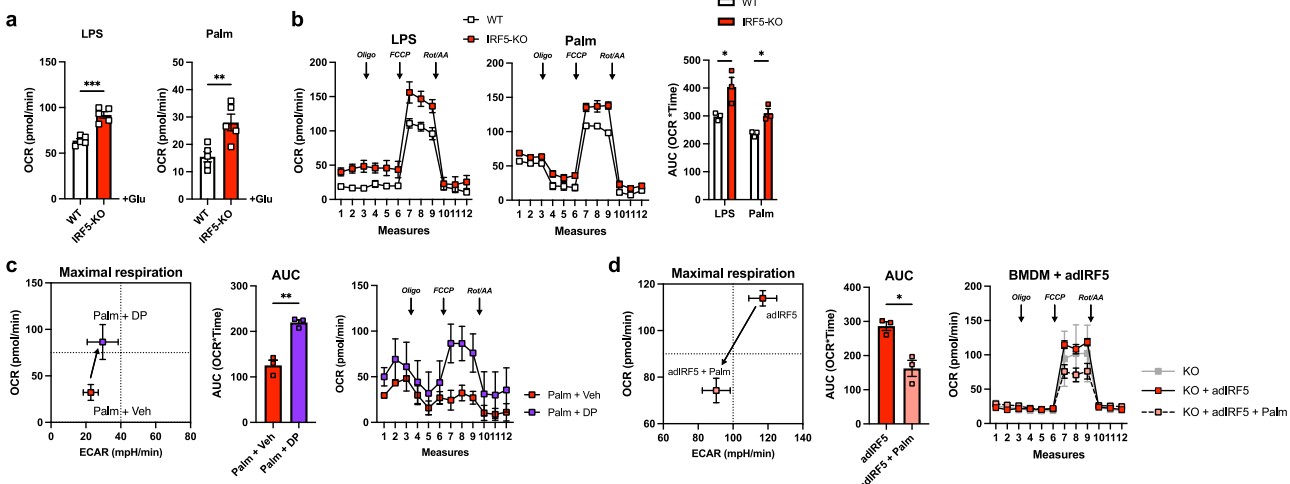

**Fig. 3 | IRF5-deficient hyperoxidative phenotype is cell intrinsic, inducible and reversible in mature bone-marrow-derived macrophages.** BMDMs from WT and IRF5-KO mice were treated with lipopolysaccharides (LPS) or palmitate (Palm) for 24 h. **a** Oxygen consumption rate (OCR) following addition of glucose (Glu) from extracellular flux analysis in Fig. S5A. (LPS ***$p = 0.0001$ and Palm **$p = 0.008$, two-tailed unpaired $t$-tests) ($n = 5$ per genotype). **b** OCR from extracellular flux analysis with following oligomycin (Oli), carbonyl cyanide 4-(trifluoromethoxy) phenylhydrazone (FCCP) and Rotenone/Antimycin A (Rot/AA) treatments (left) and area under the curve (AUC) (right) ($n = 3$ per genotype; *$p = 0.038$, and *$p = 0.013$ two-

tailed unpaired $t$-test). **c** BMDMs from C57BL/6 J mice were treated with an IRF5-decoy peptide (DP) or a vehicle (Veh) and with palmitate (Palm). Energetic plot (left) of maximal respiration from extracellular flux analysis (right) and AUC (middle) ($n = 3$ per condition, two-tailed unpaired $t$-test, **$p = 0.0022$). **d** BMDMs from IRF5-KO mice were transfected with an IRF5 adenovirus (adIRF5) and treated with palmitate. Energetic plot (left) of maximal respiration from mitochondrial stress test (right) and AUC (middle) ($n = 3$ for KO + adIRF5 and KO + adIRF5 + Palm; $n = 2$ for KO, two-tailed unpaired $t$-test between KO + adIRF5 and KO + adIRF5 + Palm, *$p = 0.01$). Data presented as mean ± SEM. Source Data file provided.

Palindromic Repeats (CRISPR)-Associated Protein (Cas)−9 linked to EGFP and under control of the *Lyz2* promoter (Fig. S8B). Transfection with gGHITM resulted in a 40 % decrease in expression (Fig. S8C). We subjected BMDM to palmitate treatment, GHITM expression decreased in response to palmitate and upon transfection (Fig. 6e, S8D, S8E). Transfection with gGHITM also decreased OCR measures, in particular at maximal respiration, in untreated and palmitate-treated cells (Fig. 6f). We then targeted *Irf5* alone (gIRF5) or co-transfected with gIRF5 and gGHITM (Fig. S8F). Extracellular flux analysis after palmitate treatment revealed that gIRF5 increased OCR, reproducing the IRF5-KO phenotype (Fig. 6h, S8G). Co-transfection with gGHITM normalised respiration to control levels (Fig. 6h, S8G). These results indicate GHITM contributes to increased oxidative respiration in IRF5-deficient macrophages.

## IRF5-GHITM regulatory axis is conserved in patients with obesity and type-2 diabetes

RNA-seq on IRF5$^+$ and IRF5$^-$ monocytes from patients with T2D revealed 3211 upregulated and 295 downregulated genes in IRF5$^+$ monocytes (Fig. 7a). Terms for mitochondrial organisation and protein localisation to mitochondria were under-represented amongst upregulated genes while downregulated genes enriched lipid catabolism terms (Fig. 7b). Additionally, *Ghitm* was consistently downregulated in IRF5$^+$ relative to IRF5$^-$ cells (Fig. 7c).

ScRNA-seq on SVF from lean and obese humans[32] confirmed previous reports that *Irf5* expression is increased with obesity and revealed a concurrent decrease in *Ghitm* expression (Fig. 7d, S9A). Cell-by-cell visualisation indicated that as cells gain expression of *Irf5*, they lose expression of *Ghitm* (Fig. 7d, S9B). We next binned cells by increasing levels of *Irf5* expression and found that as *Irf5* expression increased, the proportion of *Ghitm*$^+$ cells decreased (Fig. 7e). Correlative analyses revealed a strong negative association between *Irf5* and *Ghitm* mean expression per bin (Fig. 7f). For further analysis, we obtained WAT biopsies from a cohort of patients with obesity and sorted CD14$^+$ ATMs from subcutaneous and visceral fat depots (scATMs, vATMs) for qRT-PCR analysis. Samples were designated as IRF5$^{Hi}$ or IRF5$^{Lo}$ expressors, in which we found similar counter-

regulation of *Ghitm* in vATMs, but not in scATMs (Fig. 7g, S9C). In functional analyses, we found negative association trends between IRF5 expression and Mt Mass, mΔΨ and mΔΨ-to-mass ratio in vATMs, and mΔΨ-to-mass ratio in monocytes (Fig. 7h, S9D). These results demonstrate that the IRF5-GHITM axis is conserved in humans and may be associated with mitochondrial adaptation of ATMs and monocytes in obesity and T2D.

To evaluate the potential for transcriptional regulation, we stained monocytes from patients with T2D for IRF5 and for oxidative phosphorylation (OXPHOS) enzyme complexes (Fig. 7i). These complexes are typically anchored to the cristae structures maintained by GHITM[31,33]. Monocytes with nuclear localisation of IRF5 (Nuc) had lower OXPHOS staining density relative to those with cytoplasmic staining (Cyt). Loss of OXPHOS complex density is associated with nuclear localisation of IRF5, indicating a transcriptional mechanism. Lastly, we used the University of California Santa Cruz (UCSC) genome browser to visualise IRF5 binding regions around the *Ghitm* gene. Several IRF5 binding regions were found on and upstream of *Ghitm* (Fig. 7j). On this same resource, the expression of *Ghitm* mRNA is decreased in LPS-treated human monocyte-derived macrophages (HMDM), and this coincides with a decrease in active transcription histone mark H3K27ac. The above analyses demonstrate that IFR5 can bind to the *Ghitm* gene in humans, *Irf5* and *Ghitm* are also reciprocally regulated, indicating that IRF5's transcriptional activity may be targeted to *Ghitm* upon macrophage polarisation.

## Discussion

WAT is a key responder to caloric excess. Adaptive responses dictate disease course in metabolic syndrome, and a major determinant of tissue adaptation is the phenotype and function of ATMs. ATMs are a heterogenous population of cells ranging from regulatory to highly inflammatory, the latter contributing to the systemic metabolic decline in obesity and T2D. As sentinel cells with roles in maintaining homeostasis, the molecular mechanisms of ATM adaptation to early caloric excess remain to be fully understood. Here we demonstrate that ATMs undergo extensive IRF5-dependent energetic adaptation upon short-term caloric excess. ATM oxidative capacity is limited by

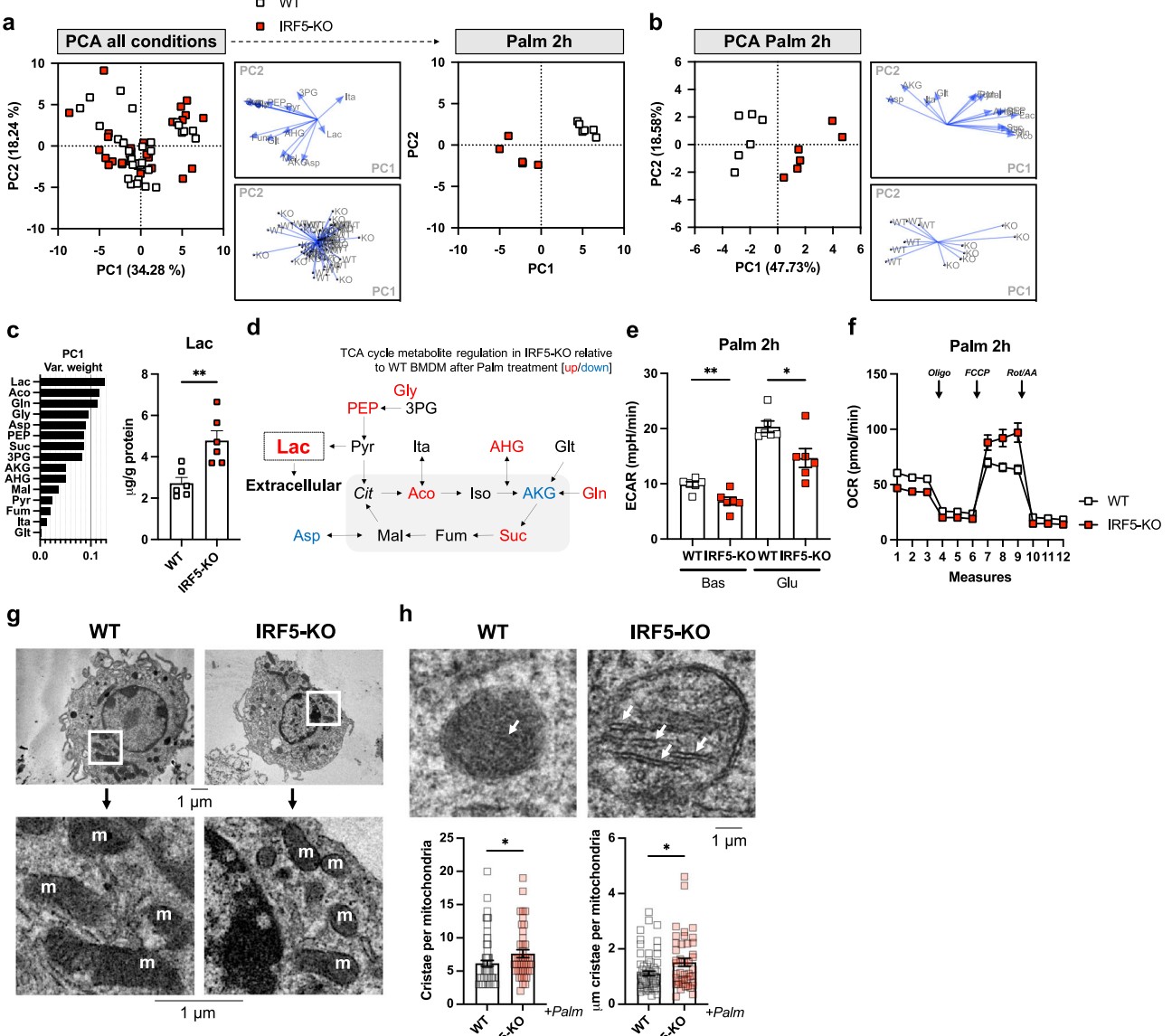

**Fig. 4 | IRF5-KO alters TCA cycle metabolite concentrations and mitochondrial structural components in macrophages in response to lipotoxicity.** Bone-marrow-derived macrophages (BMDMs) from WT and IRF5-KO mice were treated with either bacterial lipopolysaccharides (LPS) or palmitate (Palm) for 2 or 24 h. Targeted metabolomics analyses were carried out to quantify intracellular tri-carboxylic acid (TCA) cycle metabolites ($n = 5$–6 per condition) and electron microscopy was carried out to evaluate mitochondrial structural characteristics ($n = 3$ per condition). **a** Principal component analysis (PCA) on TCA cycle meta-bolites in all conditions. Palm 2 h condition separated (right). **b** PCA on TCA cycle metabolites in WT and IRF5-KO BMDMs stimulated with Palm for 2 h. **c** Variable weighting from PCA, percent variance contribution to principal component (PC)1, upon 2 h of Palm treatment. Lactate (Lac) intracellular concentration in WT and IRF5-KO BMDMs treated with Palm for 2 h ($n = 6$ per condition, **$p = 0.003$, two-tailed unpaired $t$-test). Aco aconitate, Gln glutamine, Gly glycerate, Asp aspartate, PEP phosphoenol pyruvate, Suc succinate, 3PG 3-phospho glycerate, AKG a-ketoglutarate, AHG a-hydroxyglutarate, Mal malate, Pyr pyruvate, Fum fumarate, Ita itaconate, Glt glutamate. **d** Schematic representation of metabolites with increased (red) or decreased (blue) abundance in IRF5-KO relative to WT BMDMs following treatment with Palm. Cit citrate, Iso isocitrate. **e** Extracellular acid-ification rate (ECAR) from extracellular flux analysis under Basal (Bas) and glucose-stimulated (Glu) conditions, of WT and IRF5-KO BMDMs treated with Palm for 2 h ($n = 5$ per condition, two-tailed unpaired $t$-tests, **$p = 0.006$, *$p = 0.01$). **f** Oxygen consumption rate (OCR) from extracellular flux analysis, with oligomycin (Oli), carbonyl cyanide 4-(trifluoromethoxy) phenylhydrazone (FCCP) and Rotenone/Antimycin A (Rot/AA) administration, performed on WT and IRF5-KO BMDMs, treated with Palm for 2 h ($n = 5$ per condition). **g** Electron micrograph and magnified inlet of BMDMs from WT and IRF5-KO mice after 2 h Palm treat-ment. Mitochondria are marked by 'm'. **h** Mitochondrial cristae (white arrows) in electron micrograph and length and number of cristae of BMDMs from IRF5-KO and WT mice following Palm treatment for 2 h (*$p = 0.01$, *$p = 0.04$ unpaired $t$-test). Cell were derived from $n = 3$ independent animals per genotype and indi-vidual mitochondria were analysed in each sample ($n = 26$, 23 and 20 for WT and $n = 11$, 16 and 16 for IRF5-KO). Data presented as mean ± SEM. Source Data file provided.

IRF5's transcriptional interaction with *Ghitm*, the gene coding an inner mitochondrial membrane protein that maintains mitochondrial architecture for efficient oxidative respiration. Decreased GHITM expression and loss of cristae organisation occur at an early stage of DIO and represent an IRF5-dependent mechanism that may contribute to loss of microenvironmental homeostasis and development of

insulin resistance (Fig. 8). Previous studies show that inflammation arises in WAT and is mediated by ATMs. The key implication of IRF5 in this inflammation and the development of T2D has been demonstrated[17]. *Irf5* gain-of-function risk variants have also been associated with increasing macrophage glycolytic flux[18], a cellular process that supports inflammatory effector function. A recent study

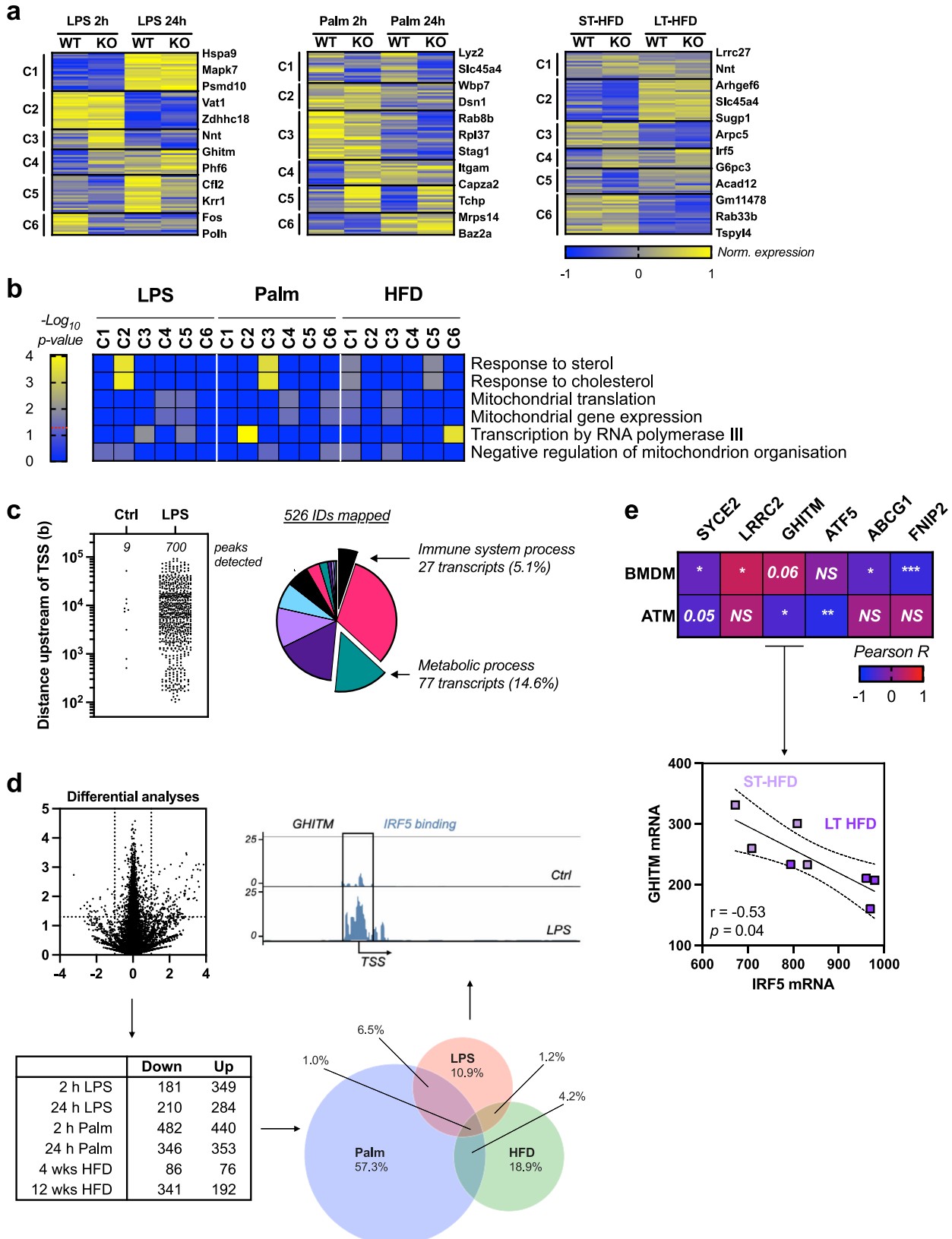

also demonstrated that IRF5 regulates airway macrophage metabolic response to viral infection[34]. Here we hypothesised that this transcription factor may have a role to play in adapting ATM metabolism in response to caloric excess. We first found that metabolically relevant genes were disproportionally represented in the IRF5-deficient transcriptome upon short-term but not long-term high-fat feeding. The

latter is enriched by inflammation-related genes. In coherence with a study by Lee et al.[16] that demonstrated immunocompromised mice developed insulin resistance upon short-term high-fat feeding, indicating that inflammation is not required for loss of glycaemic homeostasis in short-term caloric excess. A further study by Shimobayashi et al.[35] confirmed this, demonstrating that WAT was disproportionately

**Fig. 5 | IRF5 binds to and regulates expression of genes that control mitochondrial structure and metabolism in bone-marrow-derived macrophages and adipose tissue macrophages in response to metabolic stress. a** Clustering analysis on RNA sequencing from bone-marrow-derived macrophages (BMDM) from IRF5-KO and WT mice treated for 2 or 24 h with bacterial lipopolysaccharides (LPS) or palmitate (Palm) and epididymal white adipose tissue (EpiWAT) F4/80⁺ macrophages (ATMs) from IRF5-KO and WT mice following short-term (ST-) or long-term (LT-) high-fat diet (HFD). Clustering analyses was applied to genes differentially expressed between genotypes in at least one condition. **b** Gene ontology (GO) term enrichment, related to mitochondria and lipotoxicity. Genes from differentially regulated clusters in panel **a** and Fig S7A (Binomial test of gene list compared to species reference genome). **c** Publicly available chromatin immunoprecipitation (ChIP) seq of IRF5 in BMDMs treated with LPS for 120 min was

procured. Peaks of interest were determined as either at or upstream of transcription start sites (TSS). Annotated genes were subjected to gene ontology (GO) enrichment analyses. **d** Differential analysis between genotypes per treatment: 2- or 24-h treatment with LPS or Palm and in EpiWAT ATMs following ST- or LT-HFD (Wald test *p*-value <0.05, equivalent to −log₁₀ *p*-value > 1.3, and Log₂FC > 1.0). Venn diagram of differentially expressed genes between genotypes, per treatment condition. Percentage refers to proportion of genes in overlap. Gene track from ChIP-seq in **c**. of GHITM gene which overlaps all conditions and also bound by IRF5. **e** Correlative analyses of IRF5 expression and the expression of previously identified overlapping genes in **d**; and notably GHITM expression in ATMs from IRF5-competent mice fed a ST- or LT-HFD (Pearson's correlation, Pearson *r* = −0.83, two-tailed *p* = 0.009). Data presented as mean ± SEM. Source Data file provided.

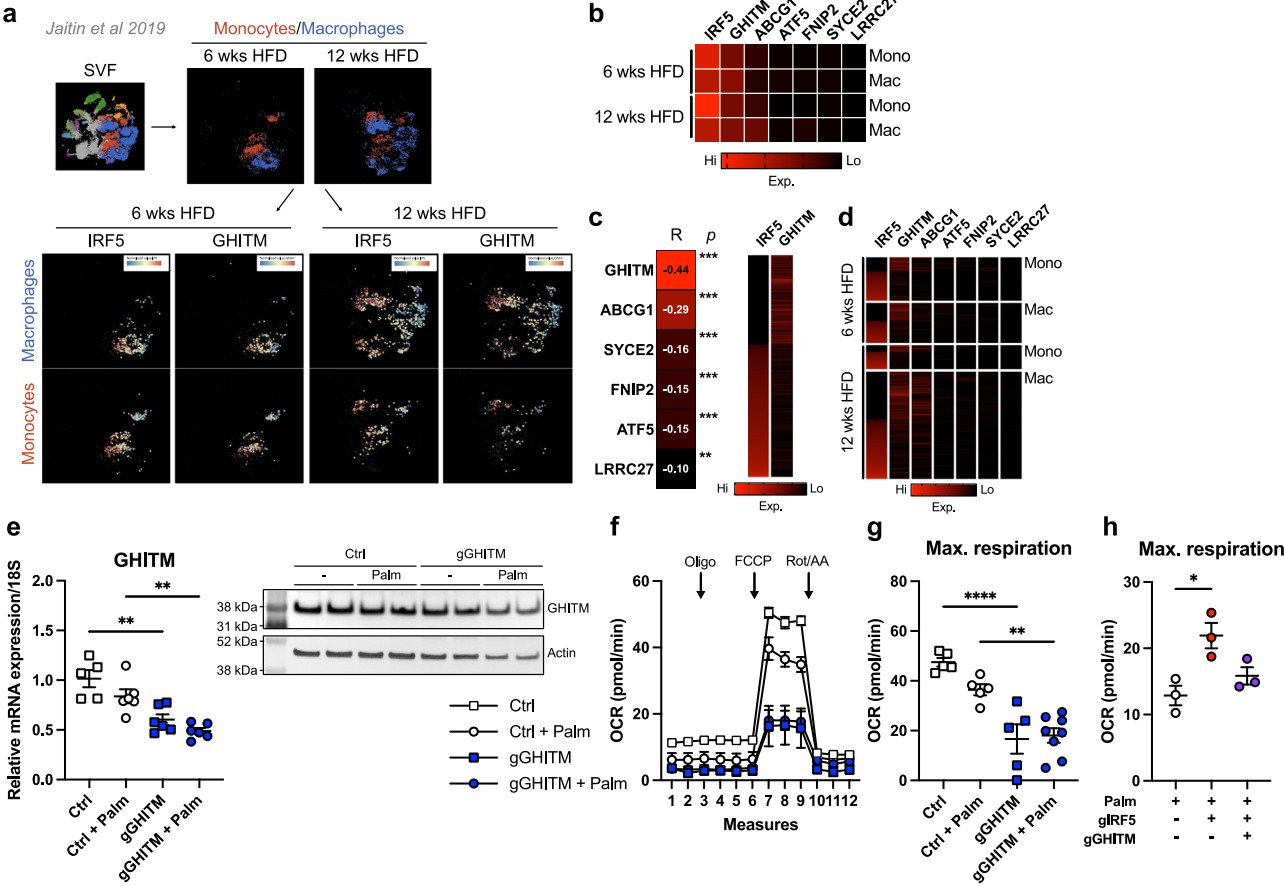

**Fig. 6 | IRF5 and GHITM are highly expressed and reciprocally regulated in epididymal white adipose tissue macrophages and monocytes. a** Single-cell RNA sequencing of the epididymal white adipose tissue (EpiWAT) stromal vascular fraction (SVF) of C57BL/6 J mice following 6 or 12 weeks of high-fat feeding (Jaitin et al., 2019). Macrophages and monocytes were identified and expression of IRF5 and of GHITM were projected onto tSNE plots per cell type and duration of high-fat feeding. **b** Heatmap of mean expression values of IRF5, GHITM, ABCG1, SYCE2, FNIP2, ATF5 and LRRC27 over time and by cell type [monocytes (Mono) or macrophages (Mac)]. **c** Correlative analyses between IRF5 expression and expression of GHITM, ABCG1, SYCE2, FNIP2, ATF5 and LRRC27 at the single-cell level (Pearson's correlation, Pearson *r*; two-tailed ***p < 0.0001 and **p = 0.004). Heatmap of IRF5 and GHITM expression, each line represents a single cell. **d** Heatmap of single-cell expression of IRF5, GHITM, ABCG1, SYCE2, FNIP2, ATF5 and LRRC27 over time and by cell type, each line represents a single cell. **e** Gene expression of GHITM in bone-marrow-derived macrophages (BMDMs) from mice with myeloid-restricted Cas9-GFP expression, treated with lipofection agent (Ctrl) or with a guide RNA (gRNA)

targeting GHITM (gGHITM) and with or without Palm treatment for 2 h (*n* = 5 for Ctrl, *n* = 6 for other conditions, one-way ANOVA. ***p = 0.0003, left *p = 0.0423, right *p = 0.0167). Western blotting against GHITM in the same experimental design, quantification and blot in Fig. S8D, S8E (*n* = 2 per condition). **f** Oxygen consumption rate (OCR) from extracellular flux analysis in BMDMs with or without Palm treatment following transfection with gGHITM or with lipofection agent alone (Ctrl). Oligomycin (Oli), carbonyl cyanide 4-(trifluoromethoxy) phenylhydrazone (FCCP) and Rotenone/Antimycin A (Rot/AA) were administered (*n* = 5 for Ctrl, Ctrl +Palm and gGHITM; *n* = 8 for gGHITM + Palm). **g** Maximal respiration from extracellular flux analysis (*n* = 5 for Ctrl, Ctrl + Palm and gGHITM; *n* = 8 for gGHITM +Palm; one-way ANOVA, ****p = 0.000065 and **p = 0.0051). **h** Maximal respiration from extracellular flux analysis on Palm-treated BMDMs following transfection with a gRNA targeting IRF5 (gIRF5), double transfection with gGHITM and gIRF5 or with lipofection agent alone (Ctrl) (*n* = 3 per condition; one-way ANOVA, *p = 0.0428). Data presented as mean ± SEM. Source Data file provided.

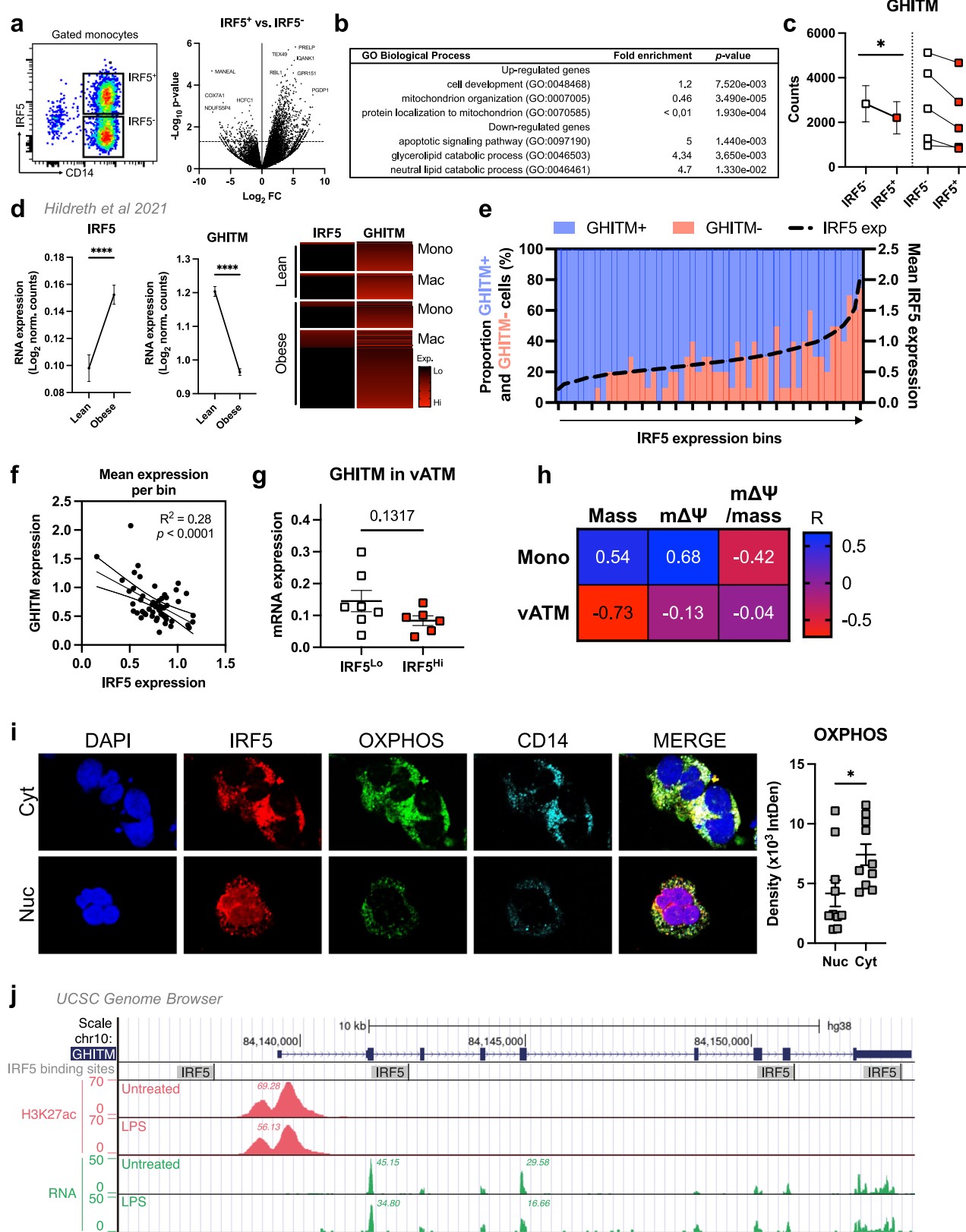

affected and that early loss of glycaemic homeostasis precedes inflammation. Our findings are supported by these studies, within 4 weeks of high-fat feeding, mice develop altered glucose homeostasis, however, without an overt IRF5-linked inflammatory signature. We did, however, demonstrate that ATMs undergo adaptation and are energetically distinct at this timepoint when compared to mice on a NCD.

ATMs reside in a lipid-rich environment and take on an overall hypermetabolic phenotype in DIO, increasing glycolysis as well as mitochondrial respiration[8]. More recent studies report specific LAM expansion on HFD, with LAMs being metabolically protective[12]. Similarly, CD11c[+], CD206[+] double-positive macrophages were found to expand on short-term HFD and are a highly oxidative population[14].

**Fig. 7 | IRF5 binds to GHITM and regulates mitochondrial activity in human monocytes and adipose tissue macrophages. a** CD14[+] Monocytes from patients with type-2 diabetes (T2D; $n = 5$) were sorted based on expression of IRF5 for RNA-seq. Differential analyses were paired by patient and carried out on IRF5[+] versus IRF5[-] monocytes ($n = 5$, Wald test $p$-value < 0.05). **b** Gene ontology (GO) term enrichment from upregulated and downregulated genes in IRF5[+] versus IRF5[-] monocytes. **c** Expression of Ghitm in IRF5[-] and IRF5[+] monocytes ($n = 5$, *$p = 0.039$, two-tailed paired $t$-test). **d** Irf5 and Ghitm counts in white adipose tissue (WAT) macrophages and monocytes, from public dataset of scRNA-seq of the stromal vascular fraction (SVF) of patients that are lean or with obesity[32] (two-tailed unpaired $t$-test, ****$p < 0.0001$). Heatmap of single-cell expression of Irf5 and Ghitm from monocytes (Mon) and macrophages (Mac), each line represents a single cell. **e** Proportion of Ghitm + (blue) and Ghitm− (red) cells in 10-cell bins by increasing Irf5 expression. **f** Correlation of Ghitm and Irf5 mean expression per bin (Pearson's correlation Pearson $R^2 = 0.28$, two-tailed $p < 0.0001$). **g** Ghitm expression in CD14[+] human visceral adipose tissue macrophages (vATMs). Samples were stratified

based on expression of Irf5 into IRF5[Lo] versus IRF5[Hi] expressors (IRF5[Lo] $n = 7$ and IRF5[Hi] $n = 6$, two-tailed unpaired $t$-test, $p = 0.13$). **h** Correlation of IRF5 MFI, JC1-Green (mitochondrial mass, Mt Mass), JC1-Red (m$\Delta\Psi$), and m$\Delta\Psi$/mass in vATM from patients with obesity and in monocytes from patients with T2D (Mono) ($n = 11$ for monocytes, $n = 9$ for ATMs, Pearson's correlation, Pearson $r$ shown, $p$-values in Fig S9D). **i** Immunofluorescence staining of IRF5 (red), oxidative phosphorylation (OXPHOS) enzyme complexes (green) and CD14 (cyan) in human monocytes from patients with T2D. Nuclei stained visualised with DAPI (blue). Samples were separated based on IRF5 localisation, either nuclear (Nuc) or cytoplasmic (Cyt). Quantification of OXPHOS staining in Nuc and Cyt samples ($n = 10$ per condition, two-tailed unpaired $t$-test, *$p = 0.0335$). **j** University of California Santa Cruz (UCSC) genome browser[51–53] (http://genome.ucsc.edu) tracks at the GHITM locus. JASPAR2020[54] tracks visualise transcription factor binding sites for IRF5. BLUEPRINT[55,56] to visualise RNA expression and H3K27 acetylation marks in LPS-treated and -untreated human monocyte-derived macrophages (session link). Data presented as mean ± SEM. Source Data file provided.

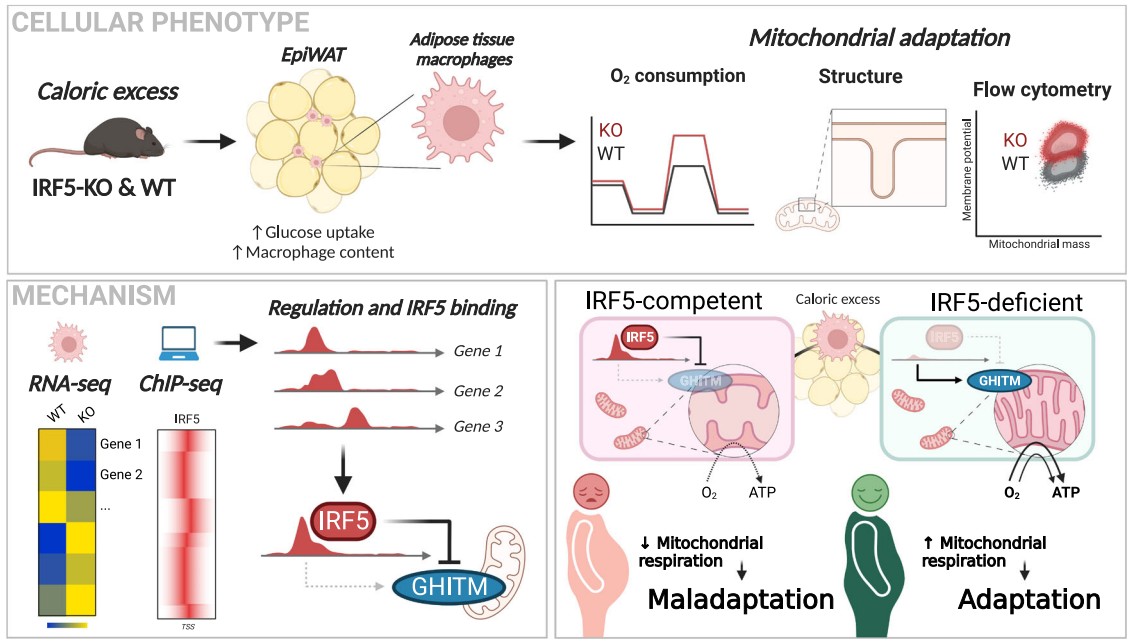

**Fig. 8 | IRF5 transcriptional interaction with the mitochondrial structural protein GHITM limits adipose tissue macrophage oxidative capacity.** This mechanism alters mitochondrial cristae structures in adipose tissue macrophages to influence tissue adaptation in diet-induced obesity. Created with BioRender.com.

While these populations of ATMs are both highly oxidative and represent a physiological adaptation to caloric excess, later engagement of glycolysis has been associated with supporting ATM inflammatory polarisation[2]. We deciphered a transcriptional mechanism that restrains cellular oxidative capacity, potentially altering microenvironmental factors and promoting greater reliance on glycolysis. IRF5-deficient macrophages have a high rate of oxygen consumption, and this is linked to a transcriptional interaction with *Ghitm*, which codes a protein that maintains the structure of OXPHOS-anchoring cristae[31]. Our finding highlights an important role for IRF5 in biasing cellular metabolism by impairing mitochondrial respiration.

We also found the TCA metabolite profile to be modified in IRF5-KO relative to WT macrophages upon stimulation. And this occurs earlier and to a greater extent in response to lipotoxicity (palmitate) than in response to bacterial stimuli (LPS). Lactate was the most important metabolite altered in IRF5 deficiency when we analysed intracellular metabolites. Its intracellular accumulation has recently been reported to be a hallmark of M2-like macrophages and supports a more tolerogenic phenotype[29]. Interestingly, increasing reports reveal intracellular lactate is itself susceptible to oxidation and can be

metabolised in mitochondria[29,36,37]. We found that decreased lactate secretion leads to its accumulation in IRF5-KO macrophages; however, we do not have direct evidence of its contribution to oxygen consumption. Moreover, the target genes that we identified have not been found to directly interact with pathways for lactate metabolism. This mechanism may contribute to the observed phenotype; however, it may also run in parallel to the structural mechanism we resolved in the IRF5-GHITM interaction. Whilst out of the scope of the current work, future investigations can focus on the mechanism by which IRF5 alters TCA cycle dynamics.

We were data-driven in resolving the current mechanism, in which we combined public datasets with our own RNA-seq to reveal that a transcriptional target of IRF5 impairs macrophage mitochondrial respiration at the early stage of glucose intolerance but prior to the onset of insulin resistance. Interestingly, this mechanism is transient, as ATMs from IRF5-KO and WT mice do not show any difference in respiratory phenotype following 12 weeks of high-fat feeding. This may be due to the function of IRF5 being more inflammatory over time or when supported by other microenvironmental cues (e.g. hypoxia, cytokines, hyperglycaemia). Such functional specificity is clearly

represented by the IRF5-deficient ATM transcriptome, which is enriched by metabolism-related genes in the short-term, and then enriched by inflammation-related genes in long-term high-fat feeding.

Previous studies found that *Ghitm* knockdown causes cristae disorganisation and mitochondrial fragmentation[31]. Mitochondrial fragmentation has been associated with inflammatory polarisation, both in response to LPS and to fatty acids[38,39]. Studies in lymphocyte lines stimulated with inflammatory cytokines and from virus-exposed monocytes also report downregulation of *Ghitm*[40,41]. These reports are in line with our current findings that loss of GHITM is associated with compromised cristae in macrophages and with increased inflammation under lipotoxic stress. We report this role for GHITM in macrophages, and monocytes, in humans and mice in response to metabolic stress. Decreased expression of *Ghitm*, and decreased ATM oxidative capacity is an early and potentially key mechanism of WAT maladaptation to caloric excess.

In summary, we deciphered a mechanism by which IRF5, a well-characterised pro-inflammatory transcription factor, alters cellular mitochondrial respiration. Having identified this mechanism to control cellular metabolism, a number of questions remain unanswered. For example, to elucidate how and through which regulatory elements IRF5 may be binding to such targets as *Ghitm*. While it is widely accepted that IRFs, target interferon-sensitive regulatory elements, it is unknown whether these response elements populate genes that regulate mitochondrial metabolism and structural components, such as GHITM. Furthermore, the specific functional contribution of GHITM downregulation to effective inflammation is unknown, for example, consequent mitochondrial fragmentation may be a source of reactive oxygen species required for bacterial killing. Lastly, despite several lines of evidence implicating IRF5 in metabolic decline associated with diet-induced obesity, the metabolic stressors that induce IRF5 expression remain unknown. Future work on the above questions will be of important insight into how this pathway can be modulated in metabolic and inflammatory diseases.

## Methods

### Ethics
All animal experiments were approved by the French ethical board (Paris-Sorbonne University, Charles Darwin N°5, 01026.02; protocols #11545, #11546, #22537, #17001), and experiments were conducted in accordance with the guidelines stated in the International Guiding Principles for Biomedical Research Involving Animals. For work with human samples, the Ethics Committee of CPP Ile-de-France approved the clinical investigations for all individuals and written informed consent was obtained from all individuals. The clinical trial principal investigator is Prof. Jean-François Gautier: jean-francois.gautier@aphp.fr. Studies were conducted in accordance with the Helsinki Declaration and were registered to a public trial registry (Clinicaltrials.gov; NCT02671864).

### Human samples and study populations
Participants were consecutively recruited, and blood samples and adipose tissue biopsies were obtained from different populations admitted to the Lariboisière and Geoffroy Saint Hilaire hospitals (Paris, France), respectively. Adipose tissue biopsies were obtained from obese subjects during bariatric surgery. Sorted and sequenced monocytes (Fig. 7a–c) were from patients with T2D aged 67–73 years old (4 male/1 female). Sorted ATMs (Fig. 7g) were from patients with obesity aged 37–54 years old (gender was anonymised for these patients). Samples analysed by cytometry (Fig. 7h) were monocytes from patients with T2D aged 45–74 years old (6 male/7 female) and ATMs from patients with obesity aged 41–59 years old (1 male/8 female). Blood samples prepared for immuno-fluorescence (Fig. 7i) were from patients with T2D aged 47–81 years old (9 male/1 female).

### Experimental animals and In vivo studies
Male C57BL/6 J mice (5–7 weeks) were purchased from Charles River. To generate mice with myeloid-specific deletion of IRF5, IRF5 flox/flox mice (C57BL/6-Irf5tm1Ppr/J; stock no. 017311) were crossed with LysM-Cre mice (B6.129P2-Lyz2tm1(cre)Ifo/J; stock no. 04781), purchased from The Jackson Laboratory. To generate mice with a restricted myeloid expression of the Cas9 endonuclease, Rosa26-Cas9KI mice (Gt(ROSA)26Sortm1.1(CAG-cas9*,-EGFP)Fezh/J; stock no. 024858, The Jackson Laboratory) were crossed with LysM-Cre mice.

Mice carrying mutated alleles were identified by PCR screening performed on genomic DNA (DNeasy Blood & Tissue Kit, Qiagen) with specific primers (Table S1). Mice were housed at 21 °C and 50% humidity, on average, on a 12 h light/dark cycle in the "Centre d'Explorations Fonctionnelles" of Sorbonne University (UMS-28). All mice used in the study were male and aged between 7 and 10 weeks old at the time of the experiment's starting point. The number of mice used per experiment is detailed in figure legends.

Mice were fed with High Fat Diet (HFD) (60% fat, D12492, Research Diets) or a normal chow diet for 4 or 12 weeks. Mice had *ad libitum* access to food and water. Mice were weighed weekly and glycaemia measured.

For the oral glucose tolerance test (GTT), mice were fasted overnight before being gavaged with glucose (2 g/kg of body weight). Tail vein blood was collected to measure glycaemia with a glucometer (Verio, One touch). For the insulin tolerance test (ITT), mice were fasted for 5 h before being i.p injected with insulin (0.5 U/kg of body weight). Glycaemia was monitored for 120 min after insulin injection.

### Organ collection and histology
Mice were sacrificed by cervical dislocation. Upon dissection, tissues were weighed. Immediately after collection, samples were either digested with collagenase, snap-frozen for further analysis or drop-fixed into 10% formalin (Sigma–Aldrich) for 24 h for histological analysis. For histological analysis, tissues were processed for dehydration, clearing and paraffin embedding with an automated carousel (Leica). Sections (6 μM thick) were stained with haematoxylin and eosin according to standard procedures. Images were acquired with a slide scanner (Zeiss Axio Scan Z1). Adipocyte diameter was measured (3 sections per mouse) with ImageJ (Fiji).

### Analysis of circulating plasma parameters
Adiponectin (Mouse Adiponectin/Acrp30 DuoSet ELISA, DY1119, R&D Systems), leptin (Mouse Leptin DuoSet ELISA, DY498-05, R&D Systems) and insulin (U-PLEX Mouse Insulin Assay, MSD) concentrations were determined by immunoassay. Plasma cytokines were quantified with LEGENDplex Mouse Inflammation kit (Biolegend) according to the manufacturer's instructions.

### Glucose uptake assay
EpiWAT explants were processed to measure glucose uptake with glucose analog 2-DG. After starvation and 2-DG uptake, explants were lysed in an extraction buffer. Lysates were processed according to the manufacturer's protocol (Glucose Uptake Fluorometric Assay Kit, MAK084, Sigma–Aldrich).

### Stromal vascular fraction
The stromal vascular fraction (SVF) containing mononuclear cells and preadipocytes was isolated from the adipose tissue after collagenase digestion. Briefly, adipose tissue biopsies were minced in collagenase solution (1 mg/ml collagenase (C6885, Sigma–Aldrich), diluted in Dulbecco's Modified Eagle Medium (DMEM) (Gibco) supplemented with 1% penicillin/streptomycin (P/S), Hepes and 2% BSA) for 20 min at 37 °C. The lysate was then passed through a 200 μM filter. After centrifugation, the resulting cell pellet was resuspended in red blood cell lysis buffer (155 mM $NH_4Cl$, 12 mM $NaHCO_3$, 0,1 mM EDTA) and passed

through a 70 µM filter. Cells were centrifuged and resuspended in FACS buffer (1× PBS supplemented with 0,5% BSA and 5 mM EDTA) for further analysis.

## Flow cytometry and cell sorting

SVF cells were prepared as described above. Blood cells were obtained from 1 ml of venous blood after red blood cells lysis and resuspended in FACS buffer.

Cells were incubated with an Fc-blocker (120-000-422, Miltenyi Biotech) for 10 min. For metabolic analysis, cells were incubated with 200 µM JC-1 (T3168, ThermoFisher Scientific) for 30 min at 37 °C. Finally, cells were stained for surface markers (Table S2) and a Live/Dead viability dye (L34957, ThermoFisher Scientific) according to the manufacturer's protocol. For intracellular lipid staining, BODIPY (D3922, ThermoFisher Scientific) was added to the surface markers antibodies mix. For IRF5 staining, cells were fixed with Foxp3-staining kit (00-5523-00, ThermoFisher Scientific) and then stained with an anti-IRF5 (10547-1-AP, Proteintech) for 1 h, and then with a secondary PE antibody (12-4739-81, ThermoFisher Scientific) for 30 min.

The acquisition was performed on a MACSQuant cytometer (Miltenyi Biotech). Cell sorting was performed on a FACSAria III (BD Biosciences). Cells were directly sorted in RLT lysis buffer supplemented with β-mercaptoethanol for RNA extraction (Qiagen). Data were analysed with FlowJo software (Tree Star). Gating strategies for analysis and cell sorting are detailed in Supplementary Methods.

Cells from the previously isolated SVF were stained for immunoselection of F4/80$^+$ or CD14$^+$ cells according to the manufacturer's protocol (MACS, Miltenyi Biotec). Cells were resuspended in MACS buffer (1× PBS supplemented with 0.5% BSA and 2 mM EDTA) containing the appropriate dilution of anti-F4/80 microbeads for murine samples (130-110-443, Miltenyi Biotec) or anti-CD14 microbeads for human samples (130-050-201, Miltenyi Biotec), for 10 min at 4 °C. Automated magnetic cell separation was performed with the Multi-MACS Cell Separator. For RNA extraction, the F4/80$^+$ cell fraction was washed and directly resuspended in RLT lysis buffer supplemented with β-mercaptoethanol (Qiagen). For metabolic flux measurements, F4/80$^+$ and F4/80$^-$ cells (180,000 cells per well, in XFe96 cell culture plates) were allowed to adhere overnight in RPMI medium supplemented with 10% FBS and 1% P/S.

## In vitro macrophage studies

**Bone-marrow-derived macrophages.** Murine bone-marrow cells were isolated from femurs and tibias. Cells were plated in DMEM (Gibco) supplemented with 10% FBS, 1% P/S and 30% L929 conditioned media and were allowed to differentiate for 8–10 days into bone-marrow-derived macrophages.

**Treatments.** Cells were treated with LPS (10 ng/ml) (L2630, Sigma–Aldrich) or palmitate (200 µM) for the appropriate time. Palmitate stock solution was prepared by dissolving sodium palmitate (P9767, Sigma–Aldrich) in 50% ethanol solution, followed by dilution in a 1% fatty acid-free albumin solution (A8806, Sigma–Aldrich).

**Decoy peptide.** Fully differentiated BMDMs were pre-treated with an IRF5-decoy peptide[27] (50 µg/ml) for 30 min, before being treated for further analysis.

**Transfection.** Fully differentiated BMDMs were transfected with IRF5 (Mm.Cas9.IRF5.1.AB, Integrated DNA Technologies) or GHITM (Mm.Cas9.GHITM.1.AA, Integrated DNA Technologies) gRNA (30 nM) complexed with lipofectamine RNAiMAX (ThermoFisher Scientific) for 48 h.

**Adenoviral transduction.** Fully differentiated BMDMs were incubated with adenovirus particles (AdIRF5 or AdGFP) for 48 h, at MOI 10.

## Immunofluorescence

After red blood cells lysis, blood cells were cytospun onto SuperFrost Plus slides. Samples were fixed in 10% formalin (Sigma–Aldrich) and then stained for CD14 (13-0149-82, Invitrogen) overnight and with the appropriate secondary antibody (Streptavidin AF 647, S32357, ThermoFisher Scientifc). Samples were then permeabilised and stained for IRF5 (10547-1-AP, Proteintech) and OXPHOS (MS604, Abcam) with the appropriate secondary antibodies (goat anti-mouse FITC (A11001) and anti-rabbit AF555 (A21428), Invitrogen). Nuclei were counterstained with Hoescht 33342 (ThermoFisher Scientific). Images were acquired with a confocal microscope (Zeiss LSM 710) and analysed with ImageJ (Fiji).

Adipose tissue sections were stained for Mac2 (CL8942AP, Cedarlanelabs) overnight and then with the appropriate secondary antibody. Nuclei were counterstained with Hoescht 33342 (ThermoFisher Scientific). Slides were scanned using Zeiss Axio Scan Z1, and Mac2 staining was quantified with Visiopharm.

## Quantitative PCR with reverse transcription

RNA was extracted from cells or tissue using RNeasy Plus Mini or Micro kit (Qiagen). Complementary DNA was synthesised with M-MLV Reverse Transcriptase kit (Promega). SYBR Green qRT-PCR reactions were performed with MESA green MasterMix (Eurogentec) and sequence-specific primers (Table S3), using QuantStudio 3 Real-Time PCR Systems (ThermoFisher Scientific). 18 S was used for normalisation to quantify relative mRNA expression levels.

## Western blotting

To extract proteins, cells were lysed in RIPA lysis buffer (Sigma), supplemented with proteases (A32955, ThermoFisher Scientific) and phosphatases inhibitors (1862495, ThermoFisher Scientific). Proteins were separated on NuPAGE 4–12% polyacrylamide gels (ThermoFisher Scientific) and then transferred onto nitrocellulose membranes. Membranes were probed with the appropriate primary (anti-GHITM, 16296-1-AP, Proteintech; anti-Actin, ab8226, Abcam) and secondary antibodies (31430 and 31460, Invitrogen) and visualised with SuperSignal West Pico Substrate (34080, ThermoFisher Scientific). Images were analysed with ImageJ (Fiji).

## Extracellular flux measurements

Real-time extracellular acidification rate (ECAR) and oxygen consumption rate (OCR) were measured using Seahorse XF24 or XFe96 extracellular flux analyser (Agilent). Briefly, cells were differentiated in XF24 or XFe96 cell culture plate (15,000–30,000 cells per well). Adipose stromal vascular cells (800,000 cells per well) were seeded in an XFe96 cell culture plate pre-treated with CellTak (Corning). F4/80$^+$ and F4/80$^-$ cells were allowed to adhere (180,000 cells per well) overnight in RPMI medium, supplemented with 10% FBS and 1% P/S. Cells were incubated in Seahorse XF base medium supplemented with either 2 mM L-glutamine, 10 mM glucose and 1 mM sodium pyruvate (pH = 7.4) for mitochondrial stress test or only 2 mM L-glutamine (pH = 7.4) for glycolysis stress test, for 1 h at 37 °C in a non-CO$_2$ incubator. For palmitate oxidation test, cells were placed in substrate limited medium (DMEM supplemented with 0.5 mM glucose, 1 mM GlutaMAX (Life Technologies), 0.5 mM carnitine and 1% FBS) for 24 h. Assay was performed in fatty acid oxidation assay buffer (111 mM NaCl, 4.7 mM KCl, 1.25 mM CaCl$_2$, 2 mM MgSO$_4$, 1.2 mM NaH$_2$PO$_4$, 2.5 mM glucose, 0.5 mM carnitine, 5 mM Hepes, pH = 7.4). Cells were pre-treated with etomoxir (40 µM) and then with palmitate (175 µM) before the assay. ECAR and OCR were measured in response to injections of either glucose (10 mM), oligomycin (1 µM) and 2-deoxyglucose (2-DG) (50 mM) for glycolysis stress test or oligomycin (1 µM), carbonyl cyanide 4-(trifluoromethoxy) phenylhydrazone (FCCP) (1 µM) and rotenone/antimycin A (0.5 µM) for mitochondrial stress and palmitate oxidation tests. All compounds were purchased from Sigma–Aldrich.

Three measurements were made under basal conditions and after each drug injection. Each measurement cycle had the following time parameters: 'mix' 3 min, 'wait' 2 min, 'measure' 3 min.

### Electron microscopy and structural analyses

BMDMs were scraped and fixed in 2 % glutaraldehyde for 2 h at 4 °C, postfixed in 1% Osmium tetroxide for 1 h at 4 °C, dehydrated, and embedded in Epon. Samples were then cut using an RMC/MTX ultramicrotome (Elexience), and ultrathin sections (60–80 nm) were mounted on copper grids, contrasted with 8% uranyl acetate and lead citrate, and observed with a Jeol 1200 EX transmission electron microscope (Jeol LTD) equipped with a MegaView II high-resolution transmission electron microscopy camera. Pictures of cell sections were taken at ×45,000 magnification. Mitochondria number per section was measured to evaluate mitochondria density. For cristae analysis, mitochondria and cristae were outlined using ImageJ (Fiji) and both the total length and number of cristae in each mitochondrion were calculated, as previously described[42]. For the analysis of mitochondria dynamics, the long and short axis of each mitochondrion, as well as their perimeter and area, were measured. From these values, aspect ratio (major axis/minor axis) and form factor (perimeter)$^2$/(4×pixArea) were calculated. TEM analyses were performed in triplicate, and a minimum of 11 images per sample were taken.

### Quantification of TCA metabolites by liquid chromatography coupled to high-resolution mass spectrometry (LC-HRMS)

**Metabolite extraction.** A volume of 170 µL of ultrapure water was added to the frozen cell pellets. At this step, 20 µL of each sample was withdrawn to further determine the total protein concentration (colorimetric quantification / Pierce BCA Protein Assay Kit, ThermoFisher Scientific). Then, 10 µL of 11 internal standards at 50 µg/mL were added to the remaining 150 µL of cell lysate: 13C5-alpha-hydroxyglutaric acid, 13C2-phosphoenolpyruvic acid, 13C4-fumaric acid, 13C3-pyruvic acid, 13C4-succinic acid (Merck), and D4-citric acid, 13C5-glutamine, D3-malic acid, 13C4,15N-aspartic-acid, 13C5-alpha-ketoglutaric acid and13C5-glutamic acid (Eurisotop), followed by a volume of 350 µL of cold methanol. The resulting samples were left on ice for 90 min. After a final centrifugation step at 20,000 × *g* for 15 min at 4 °C, supernatants were recovered and dried under a stream of nitrogen using a TurboVap instrument (ThermoFisher Scientific) and stored at −80 °C until analysis. Prior to LC-HRMS analysis, dried extracts were dissolved in 100 µL of 40 µL of chromatographic mobile phase A + 60 µL of mobile phase B (see below).

**Preparation of calibration standards.** Working solution (WS) for calibration curves and quality control solutions were prepared from two separate mother solutions (100 µg/ml in water) of each quantified compound: L-glutamic acid, L-aspartic acid, L-glutamine, succinic acid, alpha-ketoglutaric acid, trans-aconitic acid, L-(-)-malic acid, D,L-isocitric acid, D-glyceric acid, fumaric acid, citric acid, pyruvic acid, D-alpha-hydroxyglutaric acid disodium salt, D-(-)-lactic acid, D-(-)−3-phosphoglyceric acid, phosphoenolpyruvic acid and itaconic acid (all from Sigma). Several diluted solutions of calibration standard solutions (CSS) and quality control solutions (QCS) were prepared by successive two-fold dilutions of WS in ultrapure water. Then, a three-fold dilution in a BSA solution (7200 µg/mL), of each previous diluted solution (CSS1-8 and QCS1-3) was applied to prepare standards for the calibration curve (from 33.33 to 0.26 µg/mL), and quality control (from 53.33 to 1.51 µg/mL). A volume of 350 µL of cold methanol was added to each calibration curve and quality control solution and followed the metabolite extraction process.

**LC-HRMS analysis.** Targeted LC-HRMS experiments were performed using an U3000 liquid chromatography system coupled to a Q Exactive Plus mass spectrometer (ThermoFisher Scientific). The software interface was Xcalibur (version 2.1) (ThermoFisher Scientific). The mass spectrometer was externally calibrated before each analysis in ESI- polarity using the manufacturer's predefined methods and recommended calibration mixture. The LC separation was performed on a Sequant ZIC-pHILIC 5 µm, 2.1 ×150 mm column (HILIC) maintained at 45 °C (Merck, Darmstadt, Germany). Mobile phase A consisted of an aqueous buffer of 10 mM of ammonium acetate, and mobile phase B of 100% acetonitrile. Chromatographic elution was achieved with a flow rate of 200 µL/min. After injection of 10 µL of sample, elution started with an isocratic step of 2 min at 70% B, followed by a linear gradient from 70 to 40% of phase B from 2 to 7 min. The chromatographic system was then rinsed for 5 min at 0% phase B, and the run was ended with an equilibration step of 9 min. The column effluent was directly introduced into the electrospray source of the mass spectrometer, and analyses were performed in the negative ion mode. The Q Exactive Plus mass spectrometer was operated with capillary voltage set at −2.5 kV and a capillary temperature set at 350 °C. The sheath gas pressure and the auxiliary gas pressure (nitrogen) were set at 60 and 10 arbitrary units, respectively. The detection was achieved from m/z 50 to 600 in the negative ion mode and at a resolution of 70,000 at m/z 200 (full width at half maximum). All metabolites were detected as their deprotonated [M-H]- species.

Succinic acid, glyceric acid, itaconic acid, and lactic acid were detected at m/z 117.01933 (retention time (rt): 4.50 min); 105.01933 (rt 3.20 min); 129.01933 (rt 3.73 min); 89.02441 (rt 2.40 min), respectively; and quantified using 13C4-succinic acid (*m/z* 121.03251) as internal standard (ISTD). Malic acid and aconitic acid were monitored at *m/z* 133.01424 (rt 6.70 min); 173.0091 (rt 7.15 min), respectively; and quantified with D3-malic acid (m/z 136.03276). Citric acid, isocitric acid and 3-phosphoglyceric acid were monitored at m/z 191.01944 (rt 7.80); 191.01952 (rt 8.35); 184.98566 (rt 7.70 min), respectively; and quantified with D4-citric acid (*m/z* 195.04455). Pyruvic acid (*m/z* 87.00876, rt 2.25 min), aspartic acid (*m/z* 132.03023, rt 5,20 min), glutamine (*m/z* 145.06186, rt 4.75 min), glutamic acid (*m/z* 146.04588 rt 4.80 min), alpha-hydroxyglutaric acid (*m/z* 147.02989, rt 6.20 min), alpha-ketoglutaric acid (*m/z* 152.04637, rt 6.50 min), fumaric acid (*m/z* 150.03072, rt 7.05 min) and phosphoenolpyruvc acid (*m/z* 166.97509, rt 8.20 min) were all quantified with their isotopically labeled homologues (see above).

**Metabolomic data processing and quantification.** Xcalibur software was used for peak detection and integration. Metabolite quantification was performed using calibration curves established from peak area ratios between metabolites and their respective internal standard. Each metabolite amount was normalised by the protein quantity measured in each sample by BCA assay.

### RNA sequencing of BMDMs and F4/80$^+$ ATMs

After extraction, total RNA was analysed using Agilent RNA 6000 Pico Kit on the Agilent 2100 Bioanalyzer System. RNA quality was estimated based on capillary electrophoresis profiles using the RNA Integrity Number (RIN) and DV200 values. RNA-sequencing libraries were prepared using the SMARTer Stranded Total RNA-Seq Kit v2−Pico Input Mammalian (Clontech/Takara) from 10 ng of total RNA. This protocol includes a first step of RNA fragmentation using a proprietary fragmentation mix at 94 °C. The time of incubation was set up for all samples at 4 min, based on the RNA quality, and according to the manufacturer's recommendations. After fragmentation, indexed cDNA synthesis and amplification were performed, followed by a ribodepletion step using probes targeting mammalian rRNAs. PCR amplification was finally achieved on ribodepleted cDNAs, using 12 cycles estimated in accordance with the input quantity of total RNA. Library quantification and quality assessment were performed using Qubit fluorometric assay (Invitrogen) with dsDNA HS (High Sensitivity) Assay Kit and LabChip GX Touch using a High Sensitivity DNA chip (Perkin

Elmer). Libraries were then equimolarly pooled and quantified by qPCR using the KAPA library quantification kit (Roche). Sequencing was carried out using a pair-end 2 × 100 bp mode on the NovaSeq 6000 system (Illumina), targeting between 10 and 15 M clusters per sample.

STAR v2.7.3a (Spliced Transcripts Alignment To a Reference) was used to align reads to the mouse mm10 genome and generate raw counts[43]. We processed normalisation and differential expression gene analysis with DESeq2[44]. Pathway enrichment analyses were performed using clusterProfiler[45] with differentially expressed genes (abs(log2-FoldChange) > 1.3 and/or adj $p$-value < 0.05).

### RNA sequencing of IRF5[+/−] human monocytes

Complementary DNA libraries and RNA-sequencing Library preparation and Illumina sequencing were performed at the *Ecole Normale Supérieure* genomic core facility (Paris, France). Twenty nanograms of total RNA were amplified and converted to cDNA using SMART-Seq v4 Ultra Low Input RNA kit (Clontech). Afterwards an average of 150 pg of amplified cDNA was used to prepare the library following Nextera XT DNA kit (Illumina). Libraries were multiplexed by 12 on high-output flowcells. A 75 bp read sequencing was performed on a NextSeq 500 device (Illumina). A mean of 38.9 ± 8 million passing Illumina quality filter reads was obtained for each of the 12 samples.

The analyses were performed using the Eoulsan pipeline[46], including read filtering, mapping, alignment filtering, read quantification, normalisation and differential analysis: Before mapping, poly N read tails were trimmed, reads ≤40 bases were removed, and reads with quality mean ≤30 were discarded. Reads were then aligned against the hg19 genome from Ensembl version 91 using STAR (version 2.5.2b)[47]. Alignments from reads matching more than once on the reference genome were removed using Java version of samtools[47]. To compute gene expression, hg19 GTF genome annotation version 91 from Ensembl database was used. All overlapping regions between alignments and referenced exons were counted and aggregated by genes using HTSeq-count 0.5.3[48]. The sample counts were normalised using DESeq2 1.8.1[44]. Statistical treatments and differential analyses were also performed using DESeq2 1.8.1.

### Statistics

Data analysis was performed using Microsoft Excel for Mac 16.47. Statistical analysis was performed using a two-tailed $t$-test for two groups, an ordinary one-way ANOVA followed by Tukey's multiple-comparisons test for multiple groups and a two-way ANOVA followed by Bonferroni's multiple comparison test on Prism 9 for macOS (GraphPad). Correlative analyses were performed on Prism 9 for macOS, computing Pearson coefficients for normally distributed data or Spearman coefficients for non-normally distributed data (Graph-Pad). PCA analyses were carried out on Prism 9 for macOS. Trajectory-resolved clustering was carried out on the Orange (v. 3.28.0) Python toolbox[49]. Statistical approaches per data panel are detailed in figure legends.

### Public data

**Single-cell sequencing data.** Murine single-cell sequencing data from Jaitin et al.[12], were downloaded and treated using BioTuring BBrowser (v. 2.7.48)[50]. Data were filtered in BBrowser and exported in tabular format for subsequent treatment with Microsoft Excel for Mac and Prism9 for macOS. Human single-cell sequencing data were retrieved from GSE156110 raw data[32]. Clustering was performed according to the authors' instructions, and IRF5 and GHITM expressions were analysed in all macrophages and monocytes according to their lean/obese status.

**UCSC genome browser.** Gene tracks in Fig. 6e were visualised with the UCSC genome browser http://genome.ucsc.edu[51,52], using the track hubs[53]. JASPAR2020 was used to visualise transcription factor binding sites[54]. The BLUEPRINT track-set was used for RNA expression and H3K27 Ac lines[55,56]. Sample lines and tracks available through this session link / live link. The Human Dec. 2013 (GRCh38/hg38) assembly was used[57,58].

### Reporting summary

Further information on research design is available in the Nature Research Reporting Summary linked to this article.

## Data availability

Gene raw counts and raw fastq files for RNA-seq data generated in this study are available on GEO repository (www.ncbi.nlm.nih.gov/geo/). RNA-seq of IRF5[+/−] human monocytes (Fig. 7a-c) available under accession number: GSE176216 (GSM5360191-4 and GSM5360167-70 not included in study). RNA-seq of F4/80+ ATMs and BMDM from IRF5-KO and WT mice (Fig. 1a, b; Fig. 5a, b, d, e, Fig. S1A, B; Fig. S7A) available under accession numbers: GSE208648 and GSE208667, respectively. Previously published dataset analysed in this paper are from ref. 12 (Fig. 6a-d; Fig. S8A) (GSE128518) and from ref. 32 (Fig. 7d-f; Fig. S9A, B) (GSE155960). ChIP-seq data from ref. 30 (Fig. 5c, d; Fig. S7B) is available under accession number E-MTAB-2661. The GRCh38/hg38 assembly used is accessible via GenBank/RefSeq assembly accession numbers GCA_000001405.15/GCA_000001405.26. Source data are provided with this paper.

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

## Acknowledgements

Image acquisitions or cytometric analysis and sorting were done at CHIC (*Centre d'Histologie, d'Imagerie Cellulaire et de Cytométrie*, Centre de Recherche des Cordeliers UMR S 1138, Paris, France). CHIC is a member

of the SU Cell Imaging and Flow Cytometry network (LUMIC) and UPD cell imaging networks. Transmission electronic microscopy was performed at CIQLE platform (Centre d'Imagerie Quantitative Lyon-Est, Lyon, France) and we thank Elisabeth Errazuriz and Christel Cassin for their technical help. This work was supported by the France Génomique national infrastructure, funded as part of the "Investissements d'Avenir" program managed by the Agence Nationale de la Recherche (contract ANR-10-INBS-09). High-throughput sequencing has been performed by the ICGex NGS platform of the Institut Curie supported by the grant ANR-10-EQPX-03 (Equipex) from the French National Research Agency (*Agence Nationale de la Recherche; ANR; "Investissements d'Avenir"* program), by the Canceropole Ile-de-France and by the SiRIC- Curie program - SiRIC Grant "INCa-DGOS-4654". This research was supported by the French National Research Agency (*Agence Nationale de la Recherche; ANR*) ANR-JCJC grant for the MitoFLAME Project (ANR-19-CE14-0005) and by the French Society for Diabetes (*Société Francophone du Diabète; SFD*) *Allocation Exceptionnelle* to F.A. Collaboration with AstraZeneca provided support to F.A. and N.V. The European Foundation for the Study of Diabetes provided support to F.A. and J.F.G. Support was also provided by the *Commissariat à l'Energie Atomique et aux Energies Alternatives* and the MetaboHUB infrastructure (ANR-11-INBS-0010 grant) to F.C. and F.F. Grants from the European Union H2020 framework (ERC-EpiFAT 725790), ANR-PUMAS (ANR-19-CE14-0020) and INFLAMEX supported N.V. Support was provided from the *Fondation de la Recherche Médicale* (FDT202106013230) to L.O.

## Author contributions

L.O., N.V. and F.A. conceived and designed the study. L.O., T.E., R.B., J.M., D.C., Jo.C., Ju.C., C.P., A.l.H., A.k.H., F.C. and F.A. performed experiments and collected data. L.O., T.E., A.l.H., M.D., C.P., C.B., Sop.L, Son.L., J.R. and F.A. analysed data. A.H., M.D., J.M., P.L., L.G.B., S.B., Sop.L., Son.L., C.B., F.F. and F.A.C. contributed data or analysis tools. J.B.J., D.L., E.D., J.B., A.S., J.P.R., J.R. and J.F.G. provided key resources. L.O., D.L., J.B., E.D., J.P.R., J.R., J.F.G., N.V. and F.A. provided intellectual input. L.O., N.V. and F.A. wrote the manuscript.

## Competing interests

The authors declare no competing interests.

## Additional information

[1]INSERM UMR-S1151, CNRS UMR-S8253, Université Paris Cité, Institut Necker Enfants Malades, F-75015 Paris, France. [2]INSERM UMR-S1138, Université Paris Cité, Sorbonne Université, Centre de Recherche des Cordeliers, IMMEDIAB Laboratory, Paris, France. [3]CarMeN Laboratory, UMR INSERM U1060/INRA U1397, Lyon 1 University, F-69310 Pierre Bénite, France. [4]Department of Diabetes, Cochin Hospital, Assistance Publique - Hôpitaux de Paris, Université Paris Cité, Paris, France. [5]Department of Diabetes, Lariboisière Hospital, Assistance Publique - Hôpitaux de Paris, Université Paris Cité, Paris, France. [6]Université Paris-Saclay, CEA, INRAE, Département Médicaments et Technologies pour la Santé (DMTS), MetaboHUB, F-91191 Gif sur Yvette, France. [7]Service d'endocrinologie, diabétologie, maladies métaboliques, Hôpital Avicenne, 127 Rte de Stalingrad, 93 009 Bobigny, France. [8]Bioscience Metabolism, Research and Early Development, Cardiovascular, Renal and Metabolism (CVRM), BioPharmaceuticals R&D, AstraZeneca, Gothenburg, Sweden. [9]GenomiqueENS, Institut de Biologie de l'ENS (IBENS), Département de biologie, École normale supérieure, CNRS, INSERM, Université PSL, 75005 Paris, France. [10]Institut Curie Genomics of Excellence Platform, Institut Curie Research Center, PSL University, Paris, France. [11]Department of Digestive Surgery, Générale de Santé (GDS), Geoffroy Saint Hilaire Clinic, 75005 Paris, France. [12]Dasman Diabetes Institute, Kuwait, Kuwait. ✉e-mail: nicolas.venteclef@inserm.fr; fawaz.alzaid@inserm.fr

