## [Peer Review File · Nature Communications]

Early macrophage response to obesity encompasses
Interferon Regulatory Factor 5 regulated mitochondrial
architecture remodellingREVIEWER COMMENTS

Reviewer #1 (Remarks to the Author):

General comments on paper, title and abstract:

The mechanisms the authors describe are novel and interesting (and well supported by data) but the overall story of the paper is unclear and hard to follow (probably due to a lack of focus)

The title does not include the focus on obesity/diabetes at all, while this seems to be the focus of the authors in the introduction and discussion. This is confusing and makes it hard for the readers to understand what to expect and what to take home after reading.

It seems the authors are following several very distinct threads in this research, which are only connected in the discussion. What kind of link do the authors want to make to obesity/adipose tissue/diabetes? What makes the IRF5-GHITM axis and its regulation of macrophage oxidative respiration of relevance to these diseases (based on the in vivo data, relevance is questionable)? These questions should be clear throughout the whole paper.

Potentially, reconsidering the order of the figures could provide a more logical story to the reader, already from the results section onwards. Also a graphical abstract would help the readers while reading.

In vivo, the authors focus on IRF5 effect during diet-induced obesity and it is unclear whether IRF5-KO already have a primed state at baseline conditions; this is an important control to take along.

Introduction:

LAMs and MME ATMs are not mutually exclusive! The authors present them here as two different phenotypes, which is not the case. LAMs can be metabolically activated, there is no study that distinguishes them.

In general, the introduction is strongly underreferenced.

Results:

-4 weeks is extremely short to already have such a big difference in weight gain and glucose tolerance, why did the authors go for this short timing, while later on they move to longer times.

-I do not agree with 'loss of systemic insulin sensitivity'. If you look at Figure S1B, the ITT only shows the difference in basal glucose levels. This might indeed point to glucose tolerance. But the shape of the graphs do not indicate insulin insensitivity (which mostly also happens in much later stages of HFD feeding). It still shows a good response to insulin actually, besides higher basal glucose levels. So the authors cannot state that these mice lost systemic insulin sensitivity.

-Fig1D: Why did the authors take NCD and HFD together, and not look at the separate associations? This might provide more useful data.

- Why do the authors specifically focus on F4/80hi ATM? This should be better explained and IRF5 expression should be shown for all cells in HFD and NCD (e.g. in a UMAP heatmap to do this in an unbiased way). Text suggests that IRF5-KO F4/80hi ATMs show increased lipid content and glucose uptake but that is not the case when looking at Fig 1.

- In vivo, IRF5-KO had no effect on glucose homeostasis, body weight, AT weight,... This questions the relevance of the observations for this experimental disease setting and I wonder whether this should be the focus of the paper (intro and discussion suggests so).

The question that the authors should actually answer: Why is the in vitro effect not strong enough to have effect on the disease parameters in vivo. This can provide important insights and would help to increase the importance of this study. It would be good if the authors could show general measurements on adipose tissue inflammation, as this forms the link between HFD, ATM (metabolism) and systemic glucose homeostasis. Measuring crown-like structures in WAT of KO

and WT mice, cytokine secretion of WAT or inflammatory gene/protein expression in WAT could help to link in vitro and in vivo better and might result in positive in vivo data.

-p7 line 178: change comforting to confirming

-p7 conclusion of first paragraph is not supported by data; IRF5-KO does not later metabolic adaptation to caloric excess at 4w.

- epiWAT SVF also consists of a lot of pre-adipocytes, that adhere just as well as macrophages. SVFs are also used to culture and differentiate adipocytes. So I don't agree that the authors attribute the results of the SVF data only to ATMs, because you can simply not know for sure which cell populations drive the differences. Either they should have purified with F4/80, e.g. by magnetic separation, or indeed use a single cell metabolic approach, like SCENITH. I understand it will be hard for the authors to do these experiments again, but using the SVF for bulk analyses really limits the in vivo relevance of the data and does not allow to draw such hard conclusions.

- In Fig. 2A, WT does not respond to oligomycin, and overall shows extremely low OCR. I wonder if these cells are OK. Include ECAR data for these samples also.

- The discussion on AHG on page 8 does not make a lot of sense in my opinion. Glutamine is also the precursor of AHG so I wonder how the observed accumulation of glutamine can support the stated inhibition of AKG. This should be better clarified.

-p8 IRF5-KO seems to modify the destination of glycolysis end-products lactate and pyruvate. This should also be apparent when looking at ECAR so those data should be included at all points and this should be taken along in the discussion.

-IRF5 plays a role in maintaining normal TCA cycle responses to Palm and LPS -> I would state it differently. These data indicate that IRF5 deletion perturbs metabolic responses to Palm or LPS. It does not actively help to maintain?

- In Fig. 4B, after Palm treatment, the highest upregulated gene is *Lyz2*, which is actually the gene used as promoter for the Cre-recombinase. Can the author confirm with data that this (or other data) is not an artefact of using the Cre system to delete IRF5?

-Were CD14+ cells sorted from visceral or subcutaneous WAT? subcutaneous WAT in humans cannot be directly compared to eWAT in mice, it's an adipose depot that behaves totally different

-The final sentence in the results section makes me wonder what is the relevance of measuring it in diabetic/obese people? How can it help, or what does it explain in the context of this disease? It's a bit hard to follow the story.

-I would call AHG a relatively unknown metabolite. Maybe it is from the authors or in macrophages but there is a lot of research out on it in the context of cancer. There is much more known about regulation of AHG/2HG, I would say it is a bit of a stretch to say it is produced because of the low pH (which should then be mostly L2HG). Why focussing on the AHG in the discussion when not really doing something with it in the manuscript?

-CD11c is a difficult marker that is sometimes M1-like, sometimes a marker for foamy macrophages (in athero, where they are more M2-like), and for DCs. Caution with this marker.

-Fig1G: make NCD – HFD overlay to make the 2D plot even more informative.

-Fig2E: show the Seahorse kinetics data; the current OCR data suggest that this assay did not work.

Re: NCOMMS-21-17658-T Decision Letter

Dear Editors and Reviewers,

We thank the editors and reviewers for their comments and the suggestions made on our manuscript entitled “*Interferon Regulatory Factor 5 represses mitochondrial matrix protein GHITM to limit macrophage oxidative capacity in early response to obesity*”. We believe we have addressed all comments in a satisfactory manner and this has significantly improved the quality of our manuscript. In our work, we demonstrated that the pro-inflammatory transcription factor, Interferon Regulatory Factor (IRF)-5, plays a non-canonical role in orienting macrophage energetic adaptation in response to caloric excess. In a data-driven study, we deciphered a transcriptional mechanism by which IRF5 alters mitochondrial architecture to restrict macrophage oxidative capacity in the early response to obesity. This effect precedes systemic insulin resistance and is conserved in humans.

The main points we addressed are:

1. Improved overall flow and focus of the paper, including reorganizing figures, in line with reviewer and editor suggestions. We have now addressed the relevance of the described mechanisms to the pathological contexts of obesity and type-2 diabetes and propose the following new title: “*Interferon Regulatory Factor 5 represses mitochondrial matrix protein GHITM to limit macrophage oxidative capacity in early response to obesity*”.
2. We characterized the respiratory phenotype of adipose tissue macrophages (ATMs) from IRF5-KO and WT mice on a normal chow diet (NCD) to address the questions of a primed phenotype or baseline metabolic changes in IRF5-deficient macrophages.
3. We carried out more in-depth analyses of the adipose tissue phenotype upon short-term high-fat diet (ST-HFD) to link *in vitro* observations with an *in vivo* phenotype. We quantified macrophage content, crown-like structures (CLS), inflammatory gene expression, adipokine secretion, adipocyte size and glucose uptake.
4. We analysed cellular metabolic phenotype following a 12-week long-term HFD (LT-HFD) to evaluate whether cellular compensatory mechanisms are sustained to stages of insulin resistance.
5. We addressed concerns about specificity of macrophage respiratory phenotypes and counter-regulation of GHITM and IRF5 expression, in mice and humans. In line with reviewer suggestions, analyses were carried out on cells purified by magnetic separation, F4/80⁺ ATMs were then subjected to extracellular flux analyses. We acquired public single-cell sequencing data from human adipose tissue stromal vascular cells and carried out our own analyses on CD14⁺ ATMs from subcutaneous and visceral adipose tissue biopsies. In this response, we also evaluate the conserved potential of GHITM functions using the human Thp1 cell line.

We hope you find the revised version of our manuscript suitable for publication and look forward to hearing from you in due course.

Fawaz Alzaid and Nicolas Venteclef

Below is our point-by-point response, reviewer comments are in **bold text** and responses are in normal text. R#1 comments are addressed from page 2 and R#3 (new R#2) comments are addressed from page 15.

Point-by-point response to R#1

It would be good if the authors could show general measurements on adipose tissue inflammation, as this forms the link between HFD, ATM (metabolism) and systemic glucose homeostasis. Measuring crown-like structures in WAT of KO and WT mice, cytokine secretion of WAT or inflammatory gene expression in WAT could help to link *in vitro* and *in vivo* better and might result in positive *in vivo* data.

We thank the reviewer for these comments. Following ST-HFD, epididymal white adipose tissue (EpiWAT) from IRF5-KO mice had more crown-like structures (CLS) and a trend to increased macrophage content compared to WT mice (Response Figure 1A). Next, evaluating adipose phenotype, we quantified adipocyte size and frequency of small (<50 μm), medium (50-100 μm) or large (>100 μm) adipocytes. EpiWAT from IRF5-KO mice had a higher average adipocyte diameter and a higher frequency of large adipocytes compared to WT mice (Response Figure 1B). We carried out functional analysis by quantifying glucose uptake in EpiWAT explants following ST-HFD. Interestingly, EpiWAT from IRF5-KO mice had higher glucose uptake (Response Figure 1C). However, circulating insulin levels were similar between genotypes in the fed state (Response Figure 1D). Therefore, in response to the same concentration of insulin *in vivo*, EpiWAT glucose uptake is higher in IRF5-KO mice following ST-HFD (Response Figure 1D). The increased glucose uptake may also contribute to increasing adipocyte size in IRF5-KO on ST-HFD.

Response Figure 1. Analysis of adipose tissue phenotype in IRF5-KO and WT mice upon ST-HFD. Mice with a myeloid deficiency of IRF5 (IRF5-KO) and wild-type (WT) mice were placed on a 4-week short-term high-fat diet (ST-HFD). Epididymal white adipose tissue (EpiWAT) was collected for analysis.

A. EpiWAT sections stained with hematoxylin and eosin (HE) and with macrophage marker (MAC2), nuclei are visualized with DAPI. Arrows are pointing to crown-like structures (CLS). CLSs and MAC2+ cells were counted. Scale bar=100 μm .

B. Quantification of adipocyte diameter on EpiWAT sections. Frequency of small (<50 μm), medium (50-100 μm) or large (>100 μm) adipocytes.

C. Glucose uptake in EpiWAT explants of IRF5-KO and WT mice in a fed state on ST-HFD.

D. Insulin concentrations in circulation of IRF5-KO and WT mice in a fed state on ST-HFD.

We quantified expression of macrophage polarization and inflammatory markers at the tissue level and in circulation, as well as circulating markers of adipose function. Interestingly, we found an increase in F4/80 expression but no difference in inflammatory or polarization markers in EpiWAT (IL6, TNF α , CD206, CD11c) (Response Figure 2A). The local inflammatory status was mirrored systemically with no difference in plasma cytokine measures (IL10, IL6 and others; Response Figure 2B). Next, looking at adipokines and lipid markers, we found no difference in leptin nor adiponectin, nor in glycerol and triglyceride levels in plasma (Response Figure 2C).

Taken together, these data indicate that adipocyte glucose uptake is increased in the fed state, and adipocyte size is increased in IRF5-deficiency upon ST-HFD. Additionally, macrophages form CLSs more readily in IRF5-KO mice upon ST-HFD, without affecting tissue nor systemic inflammation.

Response Figure 2. Adipose tissue function and inflammatory status. Mice with a myeloid deficiency of IRF5 (IRF5-KO) and wild-type (WT) mice were placed on a 4-week short-term high-fat diet (ST-HFD). Plasma and epididymal white adipose tissue (EpiWAT) were collected for analysis.

A. Gene expression of macrophage and inflammatory markers in EpiWAT.

B. Cytokine measures in plasma.

C. Measures of adipokines (Leptin, Adiponectin), glycerol and triglycerides in circulation.

4 weeks is extremely short to already have such a big difference in weight gain and glucose tolerance, why did the authors go for this short timing, while later on they move to longer times.

We thank the reviewer for this comment. Several prior studies, including our own, allowed us to define a temporal window that distinguishes between physiological ATM adaptation and between IRF5-linked inflammation that contributes to insulin resistance in diet-induced obesity (DIO). Previous reports indicate that ST-HFD results in decreased glucose tolerance and impaired insulin sensitivity that precedes, and is independent of, inflammation (¹Lee *et al* 2011 Diabetes; ²Shimobayashi *et al* 2018 J Clin Invest). Works also identified populations of lipid-buffering macrophages arise on ST-HFD and persist through to 16 weeks of HFD alongside the expansion of inflammatory ATMs (³Brunner *et al* 2020). A further study that kinetically characterized ATMs at single-cell resolution defined *bona fide* lipid-associated macrophages (LAMs) that play physiological roles, but only substantially expand in EpiWAT following 12-16 weeks of HFD (⁴Jaitin *et al* 2019 Cell). Similarly, metabolically activated macrophages, that may phenotypically overlap with LAMs, exhibit divergent phenotypes, protective around 8 weeks of DIO then deleterious at 16-weeks (⁵Coats *et al* 2017 Cell Rep). Our previous work, with IRF5-deficient mice investigated chronic inflammation in DIO and the end-point of insulin resistance, revealed a genotype-dependent divergence in weight only occurs after 4 weeks of HFD (⁶Dalmas *et al* 2015 Nat Med).

The above kinetics range from a number of days to 16 weeks of HFD. Given the studied time-points and the emerging roles of IRF5 in macrophage metabolism (⁷Hedl *et al* 2016 Cell Rep; ⁸Albers *et al* 2021 Clin Exp Immunol) we aimed for the 4-week time-point to allow for sufficient weight gain and loss of glucose tolerance, without the frank obesity and insulin resistance observed on long-term (LT-)HFD. Thus, we

studied the influence of IRF5 on ATM metabolic and transcriptomic adaptation at 4- and 12-weeks of HFD (ST-HFD and LT-HFD). We evaluated if early ATM metabolic adaptation was IRF5-dependent, if this is sustained over time, or if previously reported roles in inflammation predominated following LT-HFD.

Indeed, with RNA-seq analysis of F4/80⁺ sorted ATMs, we demonstrated that the role of IRF5 in regulating cellular metabolism is more widely represented on ST-HFD than on LT-HFD. Differential gene expression analysis of F4/80⁺ ATMs from EpiWAT of IRF5-KO and WT mice revealed that terms relating to cellular metabolism were enriched on ST-HFD (*Cellular Metabolic Process, Metabolic Process*). Upon LT-HFD, enriched terms were those more canonically associated with immune functions, and were also phenotypically represented in our prior investigations in insulin resistance⁶ (*Immune System Process, Inflammatory Response, Tissue Remodeling*; Response Figure 3A). Enrichment of these terms was found to be functionally represented in ATMs, when evaluated by cytometric analysis of mitochondrial membrane potential (mΔΨ) and with metabolic flux analyses. ST-HFD resulted in increased mΔΨ in ATMs from IRF5-KO mice, this did not persist on LT-HFD (Response Figure 3B). Oxygen consumption rate (OCR) was higher in IRF5-KO ATMs relative to WT following ST-HFD but not following LT-HFD (Response Figure 3C). These results highlight a time-sensitive effect of IRF5 in ATMs, where early adaptation on ST-HFD affects cellular respiration, but late adaptation on LT-HFD, does not.

Response Figure 3. IRF5 influences ATM cellular metabolism upon short- but not long-term HFD.

Mice with a myeloid deficiency of IRF5 (IRF5-KO) and wild-type (WT) mice were placed on a 4-week short-term high-fat diet (ST-HFD) or a 12-week long-term (LT-)HFD. Epididymal white adipose tissue (EpiWAT) macrophages (ATMs) were analysed by flow cytometry or magnetically sorted.

A. Gene ontology (GO) term enrichment of differentially expressed genes (-log₁₀ p > 1.3) from RNA sequencing (RNA-seq) analysis of magnetically sorted F4/80⁺ ATMs from EpiWAT of IRF5-KO and WT mice following ST- or LT-HFD.

B. JC1-Red median fluorescence intensity (MFI) in F4/80^{hi} ATMs from EpiWAT of IRF5-KO and WT mice following ST- or LT-HFD.

C. Mitochondrial stress test measuring oxygen consumption rate (OCR) of ATMs from EpiWAT of IRF5-KO and WT mice following ST- or LT-HFD. Oligomycin (Oligo), carbonyl cyanide 4-(trifluoromethoxy) phenylhydrazone (FCCP) and rotenone/antimycin A (Rot/AA) were administered

EpiWAT SVF also consists of a lot of pre-adipocytes [...] SVFs are also used to culture and differentiate adipocytes. So, I don't agree that the authors attribute the results of the SVF data only to ATMs, because you can simply not know for sure which cell populations drive the differences. Either they should have purified with F4/80, e.g., by magnetic separation [...]. I understand it will be hard for the authors to do these experiments again, but using the SVF for bulk analyses really limits the *in vivo* relevance of the data and does not allow to draw such hard conclusions.

We thank the reviewer for this important comment and suggestion. In the original manuscript we observed increased oxygen consumption in the SVF fraction of EpiWAT from IRF5-KO mice relative to WT mice (Response Figure 4A). We attributed this to ATMs as we had, 1) observed increased mitochondrial $m\Delta\Psi$ by flow cytometry in F4/80^{Hi}CD11b⁺ cells (Response Figure 3B), and 2) when bone marrow-derived macrophages (BMDM), treated with palmitate or with bacterial lipopolysaccharides (LPS), resulted in a similar respiratory phenotype (Response Figure 4B). However, we appreciate the reviewer drawing our attention to this, indeed data in the original article does not allow us to rule out a contribution of other cell types to the hyperoxidative phenotype observed in bulk SVF from IRF5-KO mice. Accordingly, we used magnetic separation to purify F4/80⁺ ATMs from SVF and carry out metabolic flux analysis on both the F4/80⁺ and F4/80⁻ fractions. These analyses confirmed that F4/80⁺ cells are the major contributors to increased oxygen consumption in EpiWAT SVF from IRF5-KO mice relative to WT mice following ST-HFD, with no significant difference in the F4/80⁻ fraction (Response Figure 4C).

Response Figure 4. F4/80⁺ ATMs are main contributors to increased oxygen consumption in the stromal vascular fraction of epididymal white adipose tissue from IRF5-KO mice following short-term HFD

A. Mitochondrial stress test measuring oxygen consumption rate (OCR) basally and following addition of oligomycin (Oligo), carbonyl cyanide 4-(trifluoromethoxy) phenylhydrazone (FCCP) and rotenone/antimycin A (Rot/AA) in cells of the stromal vascular fraction (SVF) of epididymal white adipose tissue (EpiWAT) of mice with a myeloid-deficiency of IRF5 (IRF5-KO) and wild-type (WT) mice on a 4-week short-term high-fat diet (ST-HFD).

B. Mitochondrial stress test measuring OCR basally and following addition of Oligo, FCCP and Rot/AA in bone marrow-derived macrophages (BMDM) from IRF5-KO or WT mice. BMDM were pretreated with bacterial lipopolysaccharides (LPS) or palmitate (Palm).

C. Mitochondrial stress test measuring OCR basally and following addition of Oligo, FCCP and Rot/AA in magnetically sorted F4/80⁺ and F4/80⁻ cells from the EpiWAT SVF of WT and IRF5-KO mice on ST-HFD.

I do not agree with 'loss of systemic insulin sensitivity'. If you look at Figure S1B, the ITT only shows the difference in basal glucose levels. This might indeed point to glucose tolerance. But the shape of the graphs do not indicate insulin insensitivity (which mostly also happens in much later stages of HFD feeding). It still shows a good response to insulin actually, besides higher basal glucose levels. So the authors cannot state that these mice lost systemic insulin sensitivity.

We thank the reviewer for this comment. The insulin and oral glucose tolerance tests (ITT/OGTT) in Figure S1B of the original manuscript indeed show that mice respond robustly to insulin administration on ST-HFD, with increased fasting glucose relative to mice on a NCD mice. An OGTT resulted in a higher glucose peak in mice on ST-HFD relative to mice on a NCD (Response Figure 5A). As the reviewer has noted we

cannot conclude that mice have lost sensitivity to insulin from these results. Accordingly, we calculated area of the curve taking baseline measures into account⁹ and find no difference in response to ITT. The area of the curve on OGTT was however increased in mice following ST-HFD, indicating that mice have diminished glucose tolerance upon ST-HFD (Response Figure 5B).

The above findings indicate that ST-HFD results in a moderate loss of glycemic homeostasis. This is due to decreased glucose tolerance and is not associated with peripheral insulin resistance, as can be evaluated by an ITT. We thank the reviewer for bringing this to our attention, we have accordingly edited the manuscript text and the conclusions drawn from this experiment.

Response Figure 5. Short-term HFD results decreases glucose tolerance but not insulin resistance.

A. Oral glucose tolerance test (OGTT) and insulin tolerance test (ITT) of mice following normal chow diet (NCD), or short-term high-fat diet (ST-HFD).

B. Peak area of curves in A.

p7 line 178: change comforting to confirming /p7 conclusion of first paragraph is not supported by data; IRF5-KO does not later metabolic adaptation to caloric excess at 4w.

We thank the reviewer for these comments. As well as reorganizing the overall manuscript and data presentation, we have changed the phrasing of this section that refers to RNA-seq analysis to provide a more precise conclusion:

“Our previous work on IRF5 in insulin resistance found that maladaptive WAT expansion and inflammation is driven by IRF5 expression in ATMs. Given our current hypothesis that IRF5 has a distinct role to play in ATM energetic adaptation, we started this study by analyzing the ATM transcriptome from mice with a myeloid-deficiency of IRF5 (IRF5-KO) or wild-type (WT) mice. Upon a 12-week, long-term high-fat diet (LT-HFD), differentially expressed genes were associated with Inflammatory Response and Tissue Remodeling (Fig. 1A, S1A). This confirms phenotypic features that we previously reported. Positive Regulation of Metabolic Process, was a term also enriched that could indicate regulation of ATM metabolism.

The previously reported phenotype associated with IRF5-deficiency on LT-HFD started developing after 4 weeks of HFD, accordingly we decided to also investigate the ATM transcriptome at this time-point. On a short-term HFD (ST-HFD), differentially expressed genes enriched several GO terms for Metabolic Process. Interestingly, terms relating to immune function (Humoral Immune Response, Phagocytosis/Recognition) were under-represented (Fig. 1B, S1B). These results indicate that IRF5 may indeed influence ATM metabolic adaptation, in particular, in response to short-term caloric excess.”

Fig1D: Why did the authors take NCD and HFD together, and not look at the separate associations? This might provide more useful data.

We thank the reviewer for this suggestion. We placed C57BL/6J mice on a NCD or ST-HFD for 4 weeks and evaluated F4/80⁺ CD11b⁺ ATM content and expression of IRF5 in EpiWAT ATMs by flow cytometry. We found a positive correlation between expression of IRF5 and EpiWAT ATM content when both NCD and HFD are included in the analysis (Response Figure 6A). When conditions are separated, we do not observe a significant association between IRF5 expression and ATM content, but the trend remains positive on ST-HFD (Response Figure 6B). These data indicate that IRF5 protein expression by ATMs is not associated with their numbers in NCD alone or ST-HFD alone. However, IRF5 protein expression is associated with increasing ATM content in EpiWAT throughout its expansion between NCD and ST-HFD (Response Figure 6A). We confirmed this association by qPCR analysis in the whole tissue, where we observe a strong positive correlation between IRF5 and F4/80 expression (Response Figure 6C).

Response Figure 6. IRF5 expression is associated with increasing macrophage content in epididymal white adipose tissue in response to normal chow and short-term high-fat diet

- A. Association between epididymal white adipose tissue (EpiWAT) macrophage (ATM) content and IRF5 expression from ATMs of C57BL/6J mice on a normal chow diet (NCD) and a short-term high-fat diet (ST-HFD).
- B. Separate associations between EpiWAT ATM content and IRF5 expression from ATMs of C57BL/6J mice on a NCD or on a ST-HFD.
- C. Association between IRF5 and F4/80 mRNA expression in EpiWAT of C57BL/6J mice on a NCD and a ST-HFD.

Fig1G: make NCD – HFD overlay to make the 2D plot even more informative.

The figure has been updated in line with reviewer suggestion (Response Figure 7).

Response Figure 7. Overlay flow cytometry contour plot of JC1-Green (mitochondrial mass) and JC1-Red (mitochondrial membrane potential mΔΨ) fluorescence in F4/80^{Hi} adipose tissue macrophages from the epididymal fat pad of C57BL/6J mice following a normal chow diet (NCD) or a short-term high-fat diet (ST-HFD).

Fig2E: show the Seahorse kinetics data; the current OCR data suggest that this assay did not work.

We thank the reviewer for this observation. Analyses were carried out as described in the literature¹⁰ and following the manufacturer's instructions (Palmitate Oxidation Stress Test kit and FAO Substrate, Agilent). Kinetic measures represent the expected respiratory profile¹¹ (Response Figure 8A), with a decrease in OCR following oligomycin injection, an increase following FCCP and a further decrease at the third rotenone/antimycin A (Rot/AA) injection. The maximal respiration from these measures, between FCCP and Rot/AA (Response Figure 8B), is in the main figures. In the revised manuscript, we have included both, the kinetic data in Supplementary Figures and the maximal respiration in the main Figures.

Discrepancies in OCR ranges between experiments may be a reflection of the heterogenous nature of samples when working with primary culture models. So, to further confirm our finding, we carried out a supplementary experiment with a different design, where BMDM from IRF5-KO and WT mice were pretreated with palmitate then subjected to a substrate oxidation test, in which etomoxir was also included as an inhibitor. We found a similar result, following palmitate treatment, only etomoxir injection decreased OCR in BMDM from IRF5-KO mice to WT levels (Response Figure 8C). These data indicate that FAO contributes to the hyperoxidative phenotype of IRF5-deficient macrophages in response to metabolic stress.

Response Figure 8. Etomoxir normalizes high oxygen consumption rate in IRF5-KO cells to WT levels following palmitate treatment.

A. Palmitate oxidation test in cells of the stromal vascular fraction (SVF) from epididymal fat pads (EpiWAT) from IRF5-KO and WT mice following short-term high-fat diet (ST-HFD). Cells were treated with palmitate (Palm), etomoxir (ETO), glucose (Glu), 2-deoxyglucose (2-DG) and rotenone/antimycin A (Rot/AA).

B. Maximal OCR values from palmitate oxidation in A, performed on cells of the EpiWAT SVF from IRF5-KO and WT mice, after ST-HFD.

C. Maximal OCR values of IRF5-KO and WT bone marrow derived macrophages (BMDM) treated with palmitate (Palm). For this assay, cells were treated with glucose (Glu), 2-deoxy glucose (2-DG), etomoxir (ETO) and rotenone/antimycin A (Rot/AA).

In Fig. 2A, WT does not respond to oligomycin, and overall shows extremely low OCR. I wonder if these cells are OK. Include ECAR data for these samples also.

We thank the reviewer for this comment, we have included the ECAR data in this response (Response Figure 9A). In our experiment, OCR ranges from 8.2 to 92.5 pmol/min. This range is comparable to our other experiments using the mitochondrial stress test in WT cells (e.g. 10.7-110.9 pmol/min, or 7.6-100.9 pmol/min in another). The OCR in WT cells here may have appeared diminished due to the much higher OCR in cells from IRF5-KO mice, heterogeneity of the sample material (SVF) could also vary from one animal to another, potentially masking the individual responses to oligomycin. Accordingly, we present the data differently to highlight only the WT cells, and we carry out a paired comparison of means from measures before and after oligomycin (Response Figure 9B). These analyses confirm that on a sample-by-sample basis, cells of the SVF from WT mice do respond to oligomycin injection.

Alternatively, we have demonstrated that the respiratory phenotype of the SVF is mainly driven by macrophages. Previous literature¹², and our own data in BMDM, show that activated macrophages could exhibit diminished OCR or a diminished response to oligomycin. To our knowledge, this has not been demonstrated in ATMs upon short-term caloric excess. To address this, we calculated fold-decrease following oligomycin injection in ATMs of mice on a NCD or following ST-HFD. Interestingly, OCR of ATMs from mice on a NCD or following ST-HFD both decrease in response to oligomycin injection (Response Figure 9C). Lastly, with regards to cell viability, we carried out flow cytometry following the same tissue disruption methods as we use when preparing samples for metabolic flux analyses and included DAPI, as a viability marker. Viability of cells of the SVF ranges from 88-96% (Response Figure 9D).

Response Figure 9. Response to Oligomycin and viability of cells isolated from the stromal vascular fraction of epididymal fat pads.

A. ECAR values during mitochondrial stress test of cells isolated from the stromal vascular fraction (SVF) from epididymal fat pads (EpiWAT) from IRF5-KO and WT mice following short-term high-fat diet (ST-HFD). Cells were treated with oligomycin (Oligo), Carbonyl cyanide-p-trifluoromethoxyphenylhydrazone (FCCP) and Rotenone/Antimycin A (Rot/AA).

B. OCR values during mitochondrial stress test of cells isolated from the SVF from EpiWAT of WT mice following ST-HFD. Cells were treated with Oligo, FCCP and Rot/AA. Paired comparison of OCR values before and after Oligo injection.

C. OCR fold-response to Oligo of F4/80⁺ cells isolated from the EpiWAT SVF of mice upon normal chow diet (NCD) or ST-HFD.

D. Percentage of cell viability after EpiWAT SVF isolation. Dapi⁻ cells are considered viable.

IRF5 plays a role in maintaining normal TCA cycle responses to Palm and LPS -> I would state it differently. These data indicate that IRF5 deletion perturbs metabolic responses to Palm or LPS. It does not actively help to maintain?

We thank the reviewer for this comment. Indeed, we have not demonstrated that IRF5 actively participates in the maintenance TCA cycle responses, the differences may be an indirect consequence of IRF5-deficiency, or due to another parallel mechanism. Accordingly, we have reworded this section of the manuscript in accordance with the reviewer's suggestion to provide a more accurate interpretation of the result.

p8 IRF5-KO seems to modify the destination of glycolysis end-products lactate and pyruvate. This should also be apparent when looking at ECAR so those data should be included at all points and this should be taken along in the discussion.

We carried out targeted quantification of TCA cycle metabolites in BMDM from WT or IRF5-KO mice following 2 or 24 h of treatment with LPS or with palmitate (Palm). We applied principal component analysis (PCA) to determine, in an unbiased way, which conditions resulted in genotype-dependent differences in response and which metabolites were driving variation. PCA indicated that Palm resulted in a difference between genotypes following 2 h of treatment (Response Figure 10A). Here we carried out additional analysis to demonstrate that the difference was maintained when PCA was only applied to samples treated with Palm for 2 h (Response Figure 10B). When this analysis is considered, the highest contributor of variance was in-fact lactate (Response Figure 10B). When quantitative data were plotted, intracellular abundance of lactate was higher in BMDM from IRF5-KO mice relative to WT mice (Response Figure 10B).

Lactate, as a glycolysis end-product modified by IRF5-deficiency, was highlighted using a data-driven approach. Increased intracellular concentrations of lactate could be a reflection of increased production, decreased elimination, or a combination of both. In accordance with the reviewer's suggestion, we evaluated lactate release by looking at ECAR following Palm treatment. Interestingly, IRF5-KO cells had decreased ECAR, under basal conditions and following glucose injection (Response Figure 10C). Thus, the increased intracellular lactate in IRF5-KO cells may be explained, in part, by increased retention.

Intracellular lactate has recently been associated with alternative polarization, promoting a tolerogenic phenotype; and being susceptible to oxidation itself, can contribute to cellular oxygen consumption¹³. Consistent with this, we did observe increased OCR in IRF5-KO BMDM following 2 h Palm treatment (Response Figure 10D). This mechanism may contribute, parallel to the GHITM-dependent structural mechanism we reveal, to the hyperoxidative phenotype of IRF5-deficient macrophages. Further mechanistic investigation of IRF5-dependent remodeling of the TCA cycle could be an area for future investigation as it is beyond the scope of current work that characterized the IRF5-GHITM axis.

Response Figure 10. IRF5-KO alters TCA cycle metabolite concentrations in response to palmitate treatment *in vitro*.

A. Principal component analysis (PCA) on TCA cycle metabolites in WT and IRF5-KO BMDMs left untreated, or stimulated with LPS or palmitate (Palm) for 2 or 24 h. Samples only treated with Palm for 2 h are highlighted on the right.

B. Principal component analysis (PCA) on TCA cycle metabolites performed only on WT and IRF5-KO BMDMs treated with Palm for 2 h. Variable weighting from PCA, percent variance contribution to principal components (PC)1, upon 2 h of Palm stimulation. Lactate (Lac) intracellular content in WT and IRF5-KO BMDMs treated with Palm for 2 h.

C. ECAR values of WT and IRF5-KO BMDMs treated with palmitate for 2 h, at baseline (Bas) and after glucose (Glu) injection.

D. OCR values of WT and IRF5-KO BMDMs treated with palmitate for 2 h, during a mitochondrial stress test assay. Cells were administered oligomycin (Oligo), Carbonyl cyanide-p-trifluoromethoxyphenylhydrazone (FCCP) and Rotenone/Antimycin A (Rot/AA) (0

Why do the authors specifically focus on F4/80hi ATM? This should be better explained and IRF5 expression should be shown for all cells in HFD and NCD (e.g., in a UMAP heatmap to do this in an unbiased way). Text suggests that IRF5-KO F4/80hi ATMs show increased lipid content and glucose uptake but that is not the case when looking at Fig 1.

We thank the reviewer for this comment. We initially focused on F4/80^{Hi} ATMs as it was in this population that IRF5 expression was regulated upon ST-HFD (Response Figure 11A). We explain this reasoning more thoroughly in the revised manuscript. We have also included a UMAP in this response to show IRF5 expression across populations (Response Figure 11B) and conditions (Response Figure 11C).

With regards to glucose uptake and lipid content in this population, the data presented in the initial manuscript did show that lipid content had a trend to be increased in cells from IRF5-KO mice (p=0.08) with no difference in glucose uptake. Differences in these two parameters were not statistically significant between genotypes. With regards to these data, we have taken the reviewer's advice to improve focus and flow of the manuscript, including reorganization of the figures, and we have focused only on the interactions between IRF5 and mitochondrial adaptation.

Response Figure 11. IRF5 expression across the different adipose tissue macrophages (ATMs) subpopulations.

A. IRF5 median fluorescence intensity (MFI) in epididymal fat pads (EpiWAT) F4/80^{Lo} and F4/80^{Hi} macrophages, from mice upon normal chow diet (NCD) or short-term high fat diet (ST-HFD).

B. UMAP of IRF5 expression across populations, namely ATMs, F4/80^{Lo} and F4/80^{Hi} cells from the EpiWAT of mice upon NCD or ST-HFD.

C. UMAP of IRF5 expression in the ATMs across different diet conditions, namely NCD or ST-HFD.

In vivo, the authors focus on IRF5 effect during diet-induced obesity and it is unclear whether IRF5-KO already have a primed state at baseline conditions; this is an important control to take along. [and] look into the baseline metabolic changes and the possible compensatory mechanisms in the Irf5 deficient mice that allow them to maintain their metabolic state in the short term

We would like to thank the reviewer for raising this important point. Indeed, IRF5-KO phenotype at baseline is of key importance. To assess whether IRF5-KO mice already have a primed state at baseline conditions, we analysed metabolic parameters of IRF5-KO and WT mice on NCD, alongside the potential metabolic adaptations of their ATMs.

In terms of metabolic profiling, upon steady state, IRF5-KO and WT mice have similar body weight and glucose tolerance (Response Figure 12A). Considering ATM energetic adaptation, lipid content, glucose uptake, mitochondrial mass and mΔΨ did not differ with genotype (Response Figure 12B). F4/80⁺ cells isolated from the EpiWAT of IRF5-KO and WT mice display similar rates of oxygen consumption

(Response Figure 12C). Additionally, this is consistent with our *in vitro* data where untreated WT and IRF5-KO BMDMs have similar respiratory phenotypes when testing glycolysis or mitochondrial respiration (Response Figure 12D).

Response Figure 12. Systemic and cellular baseline phenotype of WT and IRF5-KO mice

A. Body weight and glucose tolerance test of WT and IRF5-KO mice on normal chow diet (NCD).

B. Bodipy, 2-NBDG, JC1-Green and JC1-Red median fluorescence intensity (MFI) in epididymal white adipose tissue (EpiWAT) F4/80^{hi} macrophages of WT and IRF5-KO mice on NCD.

C. OCR values of F4/80⁺ cells during mitochondrial stress test. Cells were isolated from the EpiWAT of WT and IRF5-KO mice on NCD. Cells were treated with oligomycin (Oligo), Carbonyl cyanide-p-trifluoromethoxyphenylhydrazone (FCCP) and Rotenone/Antimycin A (Rot/AA).

D. OCR values of WT and IRF5-KO bone marrow-derived macrophages (BMDMs) during mitochondrial stress test. Cells were treated with Oligo, FCCP and Rot/AA. ECAR values of WT and IRF5-KO BMDMs during glycolysis assay. Cells were treated with glucose (Glu), Oligo and 2-deoxy glucose (2-DG).

In Fig. 4B, after Palm treatment, the highest upregulated gene is *Lyz2*, which is actually the gene used as promoter for the Cre-recombinase. Can the author confirm with data that this (or other data) is not an artefact of using the Cre system to delete IRF5?

We thank the reviewer for pointing this out. *Lyz2* is one of the most downregulated genes, and this is a direct result of using the LysMCre model to target IRF5 in myeloid cells. In the LysMCre model, the Cre recombinase is inserted into the first coding ATG of the *Lyz2* gene. This abolishes endogenous *Lyz2* gene expression and function, and places Cre expression under the control of the endogenous *Lyz2* promoter/enhancer elements¹⁴. The downregulation of *Lyz2* expression does not affect macrophage phenotype, and numerous other examples in the literature have applied this model^{15,16}. Indeed, in previously published work from our group, the myeloid-specific deletion of the co-repressor GPS2 using the LysMCre model is associated with a pro-inflammatory gene signature in macrophages¹⁷. This inflammatory phenotype is opposite to what we and others have reported with myeloid-specific IRF5-deficiency using the same Cre source^{6,18}. Moreover, in the study of LysMCre restricted GPS2, *Lyz2* expression is also downregulated (Response Figure 13A), similar to what we observe across conditions in our current study (Response Figure 13B and C).

Response Figure 13. Lyz2 downregulation in the LysMCre model

A. Lyz2 RNA counts in WT and GPS2-KO bone marrow derived macrophages (BMDMs). Data originated from Fan et al., Nature Medicine (2016)

B. Lyz2 RNA counts in WT and IRF5-KO BMDMs treated with LPS, Palmitate (Palm) or vehicle for 2 or 24 hours, or left untreated.

C. Lyz2 RNA counts in epididymal white adipose tissue F4/80+ cells of WT and IRF5-KO mice on short or long-term high fat diet (ST- and LT-HFD)

CD11c is a difficult marker that is sometimes M1-like, sometimes a marker for foamy macrophages (in athero, where they are more M2-like), and for DCs. Caution with this marker.

We thank the reviewer for raising this point and have taken input into consideration. CD11c is indeed a commonly used marker for dendritic cells. Here, we particularly focused on this marker to identify inflammatory macrophages associated with IRF5 expression. IRF5 was previously defined as a marker of M1-like macrophages *in vivo*¹⁹ and has been more recently identified as a transcriptional regulator of CD11c expression, including in the context of atherosclerosis²⁰. IRF5 binds to the *Itgax* (encoding CD11c) locus in response to LPS and controls its expression (Response Figure 14)²⁰. IRF5 also drives differentiation of monocytes into CD11c⁺ inflammatory macrophages, this has been demonstrated in different tissues and pathological conditions, including atherosclerosis and intestinal inflammation^{20,21}.

These reports demonstrate that the link between IRF5 and CD11c expression is well established. However, we do acknowledge the reviewers concern, that the link between CD11c and inflammatory macrophages may be context-dependent. In particular that CD11c can be expressed by foam cells, not typically considered inflammatory, and has also been reported to be expressed on CD206⁺ ATMs in the context of DIO. Interestingly, CD11c⁺ CD206⁺ ATMs have been reported to have increased lipid-buffering and oxidative capacities in DIO, indicating they may be beneficial³.

Given the above discussion and scope of the current manuscript, investigating interaction of IRF5 with mitochondrial metabolism in ATMs, we have decided to follow the reviewer's advice and limit the use and discussion of CD11c as a marker for pro-inflammatory/M1-like macrophages in the revised manuscript. We believe this also improves the focus of the manuscript.

Response Figure 14. CD11c gene is a direct target of IRF5 in macrophages (adapted from Seneviratne *et al.*, *Circulation* (2017))
A. IRF5 binding sites from unstimulated and LPS treated bone marrow macrophages (BMDMs), with IRF5 binding to the ITGAX gene loci.
B. Itgax gene expression in BMDM from ApoE^{-/-} mice with or without a genetic deletion of IRF5 (ApoE^{-/-} Irf5^{-/-}), treated with control media or with LPS to induce IRF5 binding to the chromatin.

Were CD14⁺ cells sorted from visceral or subcutaneous WAT? subcutaneous WAT in humans cannot be directly compared to eWAT in mice, it's an adipose depot that behaves totally different

We thank the reviewer for this comment, we have now carried out analyses on ATMs from both subcutaneous (sc) and visceral (v) white adipose tissue (WAT). Biopsies were collected from a cohort of obese patients having undergone bariatric surgery, CD14⁺ cells were magnetically sorted and processed for qRT-PCR analysis of IRF5 and GHITM expression. When samples were sorted into IRF5^{Hi} versus IRF5^{Lo} expressors, we found a trend to decreased GHITM expression in IRF5^{Hi} vATMs (Response Figure 15A) and this trend was not observed in scATMs (Response Figure 15B).

Response Figure 15. IRF5 and GHITM expression in human adipose tissue macrophages
A. IRF5 and GHITM expression in visceral adipose tissue CD14⁺ cells (vATMs) from obese patients
B. IRF5 and GHITM expression in in subcutaneous adipose tissue CD14⁺ cells (scATM) from obese patients. Patients were stratified into two groups according to IRF5 expression (IRF5^{Lo} and IRF5^{Hi}).

The discussion on AHG on page 8 does not make a lot of sense in my opinion. Glutamine is also the precursor of AHG so I wonder how the observed accumulation of glutamine can support the stated inhibition of AKG. This should be better clarified. [and...]

I would call AHG a relatively unknown metabolite. Maybe it is for the authors or in macrophages but there is a lot of research out on it in the context of cancer. There is much more known about regulation of AHG/2HG, I would say it is a bit of a stretch to say it is produced because of the low pH (which should then be mostly L2HG). Why focussing on the AHG in the discussion when not really doing something with it in the manuscript?

We thank the reviewer for this comment, we have reanalyzed data from TCA cycle metabolite quantification and this is presented and discussed differently in the revised manuscript.

Point-by-point response to R#3 (new R#2)

According to Serbulea V. et al. cited by the authors, F4/80^{hi} harbor highly activated bioenergetics which rationalize the strategy used by the authors to only study F4/80^{hi} ATMs. However I think that this should be clarify in the text. In addition, have the authors analyzed the F4/80^{lo} in terms of lipid content, glucose uptake, Mt mass and activity? It would be really interesting to check the difference between resident and infiltrating ATMs considering that the authors make a point to look at monocyte-derived ATMs.

We thank the reviewer for this comment. We have taken this into consideration and clarified rationale for focusing on F4/80^{hi} adipose tissue macrophages (ATMs) in the revised manuscript. Our findings are in agreement with Serbulea *et al's* report that F4/80^{hi} ATMs are energetically distinct²². We also show that IRF5 was expressed in both F4/80^{Lo} and F4/80^{Hi} ATMs, however it was only upregulated in the F4/80^{Hi} subpopulation in response to our model of 4-week short-term (ST-) high-fat diet (HFD) (Response Figure 16A). We confirmed findings from supervised analyses with a UMAP of ATMs overlaying IRF5 expression across populations and upon ST-HFD (Response Figure 16B).

With regards to the metabolic adaptation, lipid content did not vary in F4/80^{Lo} ATMs but was higher in F4/80^{Hi} ATMs following ST-HFD (Response Figure 16C). Mitochondrial mass (Mt Mass) was increased in both F4/80^{Lo} and F4/80^{Hi} ATMs following ST-HFD (Response Figure 16D). Interestingly, mitochondrial membrane potential ($m\Delta\Psi$) was specifically decreased in F4/80^{Hi} ATMs following ST-HFD (Response Figure 16D). Analyzing both populations shows that F4/80^{Hi} ATMs have increased IRF5 expression and enhanced lipid content, yet have decreased $m\Delta\Psi$ in response to ST-HFD. Increased Mt Mass is a feature of both subpopulations.

IRF5-KO mice on a short term HFD do not display any difference in systemic metabolism (based on ITT, GTT, PTT) but their ATMs show increased mitochondrial activity and activity-to-mass ratio. On the other hand, the authors have shown that IRF5-KO mice have marked metabolic differences compared to WT upon long term HFD. This reviewer wonders whether the “metabolic adaptation” observed is actually protective as long as ATMs can handle the overload of nutrients (lipids or glucose) but could it be the proportion of resident vs. recruited ATMs with different bioenergetics that impacts the overall cellular metabolism of ATMs? What is the proportion of F4/80^{hi} vs F4/80^{lo} in short and long term HFD in IRF5-KO vs WT? The RNAseq was done in the whole ATM population and confirms the transient metabolic shift. It would be interesting to at least look at some of the readouts utilized at 4 weeks (lipid content, glucose uptake, Mt mass and activity) after long term HFD when IRF5-KO show differences in whole body metabolism.

We thank the reviewer for this insightful comment. To evaluate whether altered proportions of F4/80^{Lo} or F4/80^{Hi} ATMs are a feature associated with the protective phenotype in IRF5-deficiency, we analysed proportions in fat pads from WT and IRF5-KO mice following ST-HFD and a 12-week long-term (LT)-HFD. We found no genotype-dependent difference in proportions of F4/80^{Lo} or F4/80^{Hi} macrophages upon ST- or LT-HFD (Response Figure 17A). When considering proportions of the parent population (all F4/80+ ATMs), IRF5-KO mice had a higher proportion of ATMs, compared to WT mice following LT-HFD, but not at the earlier stage of ST-HFD (Response Figure 17B).

RNA-seq on F4/80+ ATMs from fat pads of WT and IRF5-KO mice shows that ATMs undergo an IRF5-dependent metabolic shift upon ST-HFD. Differentially expressed genes enriched terms relating to metabolism (*Cellular Metabolic Process, Primary Metabolic Process*), whereas immune-related terms were under-represented. As the reviewer has noted, this shift is transient and following a LT-HFD differentially expressed genes enrich terms more closely related to immune functions (*Inflammatory Response, Tissue Remodeling*), and more typically associated with IRF5 action^{6,19} (Response Figure 17C).

These results indicate that IRF5-deficiency does not promote predominance of one ATM subpopulation (F4/80^{Lo} or F4/80^{Hi}) over another, however it does promote an increase in overall ATM proportions following LT-HFD. Additionally, the cellular energetic phenotype we describe is pronounced in F4/80^{Hi} ATMs and does not influence their proportions in tissue upon ST-HFD. With respect to ST- versus LT-HFD, transcriptomic analysis indicates separate and time-sensitive mechanisms may be at play in IRF5-deficiency. These could represent an early mechanism improving capacity for physiological adaptation and a late mechanism that counters inflammatory polarization and improves insulin sensitivity.

As the reviewer suggested, to establish whether the cellular energetic phenotype is indeed transient on ST-HFD, or sustained on LT-HFD in support of primary anti-inflammatory and tissue-remodeling functions, we carried out extracellular flux analysis on ATMs following ST- and LT-HFD in IRF5-KO and WT mice. We found the hyperoxidative phenotype, associated with IRF5-deficiency, to only be present following ST-HFD. ATM oxygen consumption was normalized to WT levels upon LT-HFD (Response Figure 17D). These results, supported by RNA-seq data, indicate a transient role for IRF5 in adapting ATM metabolism.

Response Figure 17. IRF5-dependent metabolic adaptation of adipose tissue macrophages on short-term high-fat diet is transient

A. Proportions of F4/80^{Lo} and F4/80^{Hi} adipose tissue macrophages (ATMs) from fat pads of IRF5-KO or WT mice following short-term (ST-) or long-term (LT-) high-fat diet (HFD).

B. Proportion of ATMs (both F4/80^{Lo} and F4/80^{Hi}) amongst Live CD45⁺ and lineage (CD3, CD19 Lin) negative cells from fat pads of IRF5-KO and WT mice following ST- or LT-HFD.

C. Gene ontology (GO) term enrichment of differentially expressed genes ($-\log_{10} p > 1.3$) from RNA sequencing (RNA-seq) analysis of magnetically sorted F4/80⁺ ATMs from fat pads of IRF5-KO and WT mice following ST- or LT-HFD.

D. Mitochondrial stress test measuring oxygen consumption rate (OCR) of ATMs from fat pads of IRF5-KO and WT mice following ST- or LT-HFD. Oligomycin (Oligo), carbonyl cyanide 4-(trifluoromethoxy) phenylhydrazone (FCCP) and rotenone/antimycin A (Rot/AA) were administered.

To support the above, we evaluated lipid uptake, mitochondrial mass (Mt Mass) and membrane potential ($m\Delta\Psi$) of F4/80^{Lo} and F4/80^{Hi} ATMs following ST-HFD and LT-HFD. Upon ST-HFD, lipid content of F4/80^{Hi} ATMs had an increasing trend in IRF5-KO mice relative WT mice. On LT-HFD, this trend extended to the whole ATM population and F4/80^{Lo} ATMs, and was significant in F4/80^{Hi} ATMs (Response Figure 18A). Mt Mass was the least variable parameter between the genotypes, only on ST-HFD did F4/80^{Lo} ATMs from IRF5-KO mice have increased Mt Mass relative to WT mice (Response Figure 18B). Mitochondrial membrane potential ($m\Delta\Psi$) was increased in the whole ATM population from IRF5-KO mice upon ST-HFD, this increase was only reflected by F4/80^{Hi} ATMs (Response Figure 18C). Upon LT-HFD, ATM $m\Delta\Psi$ from IRF5-KO mice was normalized to WT levels in all populations.

Taken together, these data confirm that F4/80^{Hi} ATMs are energetically distinct and IRF5-deficiency impacts mitochondrial adaptation upon ST- but not LT-HFD. This transient effect is pronounced in F4/80^{Hi} ATMs; however, it can be detected in the whole ATM parent population by extracellular flux analysis (Response Figure 17D) and by flow cytometry (Response Figure 18C).

Response Figure 18. Energetic adaptation of F4/80^{Lo} and F4/80^{Hi} ATMs in IRF5-KO and WT mice following a short- or long-term high-fat diet. Mice with a myeloid deficiency of IRF5 (IRF5-KO) and wild-type (WT) mice were placed on a 4-week short-term (ST-) high-fat diet (HFD) or a 12-week long-term (LT-) HFD. Epididymal white adipose tissue (EpiWAT) macrophages (ATMs) were analysed by flow cytometry. Analyses were carried out on the ATM population (F4/80^{Lo+Hi}) or on separated F4/80^{Lo} and F4/80^{Hi} ATMs.

A. Lipid content was measured by BODIPY median fluorescence intensity (MFI).

B. Mitochondrial mass (Mt Mass) was measured by JC1 Green MFI.

C. Mitochondrial membrane potential (mΔΨ) was measured by JC1 Red MFI.

1/2: The differences between what has been described, particularly by the group of Luke O’neil, in terms of the TCA cycle break and the role of itaconate and the data presented here is very interesting. It would be good for the authors to go a bit further in their discussion to explain these differences.

We thank the reviewer for this comment. Luke O’Neill’s lab demonstrated that itaconate accumulates in lipopolysaccharide (LPS)-stimulated macrophages^{23,24} and can act to modulate inflammation. We replicated the previously described kinetics that show an accumulation of itaconate following LPS stimulation (Response Figure 19A). Itaconate concentrations are not altered by IRF5-deficiency, indicating that mechanisms that regulate itaconate levels are independent of IRF5. Interestingly, when considering all conditions (2 or 24 h treatment with LPS or palmitate), principal component analysis (PCA) revealed that TCA cycle metabolite concentrations only separate across genotypes in response to 2h palmitate treatment (Response Figure 19B). We confirmed this by re-applying PCA only to these samples and found lactate (Lac) to be the biggest contributor to variance (> 10 % variable weight) (Response Figure 19C) and itaconate was amongst the lowest contributors (< 2 % variable weight). Similar to the LPS response, these data indicate that itaconate levels are not regulated by, or do not contribute to, the IRF5-dependent remodeling of TCA-cycle metabolites in BMDM.

Many reports on immunomodulatory roles of itaconate centered around the IL1 axis of inflammation, as opposed to the type-1 interferon axis that is regulated by IRF5. Studies that characterized roles of itaconate have reported on the functional specificity towards IL1B^{25,26}. IL1B is sensitive to succinate accumulation, as a consequence of itaconate-mediated inhibition of succinate dehydrogenase, but other inflammatory mediators (i.e. TNF) were not^{25,26}. One study did report an NRF2-dependent feedback loop between type-1 interferon signals (IFNB) and itaconate in Poly I:C-stimulated macrophages, however this has not been described in the case of LPS-priming or with metabolic stimuli (i.e. Palm)²⁴. Taken together, there is no indication of a direct interaction between IRF5 and itaconate-mediated control of macrophage polarization in the context we have studied.

Response Figure 19. IRF5-KO alters TCA cycle metabolite concentrations in response to palmitate treatment *in vitro*.

A. Itaconate concentration in bone marrow-derived macrophages (BMDM) from IRF5-KO and WT mice following LPS treatments
B. Principal component analysis (PCA) on TCA cycle metabolites in WT and IRF5-KO BMDMs left untreated, or stimulated with LPS or palmitate (Palm) for 2 or 24 h. Samples only treated with Palm for 2 h are separated on the right.

C. PCA on TCA cycle metabolites performed only on WT and IRF5-KO BMDMs treated with Palm for 2 h. Variable weighting from PCA, percent variance contribution to principal components (PC)1, upon 2 h of Palm stimulation.

D. Intracellular lactate (Lac) content in WT and IRF5-KO BMDMs treated with Palm for 2 h.

E. ECAR values of WT and IRF5-KO BMDMs treated with palmitate for 2 h, at baseline (Bas) and after glucose (Glu) injection.

As both R#1 and R#3 (new R#2) suggested, we refocused the discussion of this section on IRF5-deficiency and TCA cycle remodeling to improve interpretation of the results. The above analyses highlighted that lactate is a main metabolite modified by IRF5-deficiency. Lactate had a higher intracellular concentration in BMDM from IRF5-KO mice, compared to WT mice (Response Figure 19D). Typically, lactate is secreted into the microenvironment following its production, to evaluate this, we looked at extracellular

acidification rate (ECAR). IRF5-deficient cells had lower ECAR than IRF5-competent cells under basal and glucose-stimulated conditions (Response Figure 19E). Thus, increased intracellular lactate in BMDM from IRF5-KO mice may be explained, in part, by its increased retention. Intracellular lactate has recently been associated with alternative polarization, promoting a tolerogenic phenotype; and being susceptible to oxidation itself, can contribute to cellular oxygen consumption¹³. This mechanism may contribute, parallel to the GHITM-dependent mechanism, to the hyperoxidative phenotype of IRF5-deficient macrophages.

2/2: Could it be a difference in fuel availability or fuel utilization (availability of high levels of FFA and glucose vs metabolism of FFA and glucose).

In terms of fuel availability and utilization that the reviewer draws attention to, we found no difference in glycaemia, plasma glycerol or triglycerides of IRF5-KO and WT mice following ST-HFD (Response Figure 20A). Indicating, that ATMs are exposed to the same systemic concentrations of extracellular substrates in IRF5-KO and WT mice. To functionally evaluate fuel utilization, we carried out a palmitate oxidation test in cells of the stromal vascular fraction (SVF) of IRF5-KO and WT mice following ST-HFD. This test evaluates fatty acid oxidation (FAO) capacity and reliance on Carnitine Palmitoyltransferase (CPT)-1, a rate-limiting enzyme for FAO. Similar to the mitochondrial stress test, palmitate loading resulted in a higher oxygen consumption rate (OCR) in the SVF of IRF5-KO mice relative to WT. With the use of Etomoxir to inhibit CPT1, the higher OCR was normalized to WT levels (Response Figure 20B). Effects were most pronounced at maximal respiration (Response Figure 20C). These data indicate a difference in fuel utilization in cells of the SVF from IRF5-KO mice compared to WT, notably a higher capacity for FAO. To confirm this difference in macrophages, we carried out a supplementary experiment where BMDM were pretreated with palmitate then subjected to a substrate oxidation test. We found a similar result, only etomoxir decreased OCR in BMDM from IRF5-KO mice to WT levels (Response Figure 20D).

Response Figure 20. Cells from IRF5-KO mice have increased capacity to oxidize lipids. (legend continues on next page)

A. Mice with a myeloid deficiency of IRF5 (IRF5-KO) and wild-type (WT) mice were placed on a 4-week short-term high-fat diet (ST-HFD). Random glycaemia was measured. Plasma was collected to measure Glycerol and Triglyceride concentrations.

Response Figure 20. Cells from IRF5-KO mice have increased capacity to oxidize lipids (legend continued)

B. Palmitate oxidation test in cells of the stromal vascular fraction (SVF) from epididymal fat pads (EpiWAT) from IRF5-KO and WT mice following ST-HFD. Cells were treated with palmitate (Palm), etomoxir (ETO), glucose (Glu), 2-deoxyglucose (2-DG) and rotenone/antimycin A (Rot/AA).

C. Maximal OCR values from palmitate oxidation in **B**, performed on cells of the EpiWAT SVF from IRF5-KO and WT mice, after ST-HFD.

D. OCR values of IRF5-KO and WT bone marrow-derived macrophages (BMDM) treated with palmitate (Palm). For this assay, cells were treated with glucose (Glu), 2-deoxy glucose (2-DG), etomoxir (ETO) and rotenone/antimycin A (Rot/AA).

The translational approach is remarkable but it might be good to at least silence GHITM to confirm its role in mitochondrial biogenesis/inflammation in human macrophages.

We thank the reviewer for this suggestion. In the manuscript we demonstrated that coregulation of IRF5 and GHITM are conserved in human myeloid cells. We used primary monocytes from patients with type-2 diabetes (T2D; Response Figure 21A) and publicly available scRNA-seq from human ATMs²⁷ (Response Figure 21B). Functionally, IRF5 expression was associated with decreased mΔΨ-to-Mt mass ratio in monocytes and ATMs, in patients with T2D or obesity, respectively (Response Figure 21C). Using the University of California Santa Cruz (UCSC) genome browser, we also found conserved potential for transcriptional interaction by mapping IRF5 binding sites to the GHITM locus (Response Figure 21D). A transcriptional mechanism was also supported by staining in monocytes, where a decreased signal for oxidative phosphorylation (OXPHOS) complexes was found when IRF5 has a predominantly nuclear (Nuc) localization, compared to predominantly cytoplasmic (Cyt) (Response Figure 21E).

Response Figure 21. The IRF5-GHITM axis is conserved in human monocytes and adipose tissue macrophages (legend continues on next page)

A. CD14⁺ Monocytes from patients with type-2 diabetes (T2D; n=5) were sorted based on their expression of IRF5 and subjected to RNA sequencing (RNA-seq). Expression of GHITM mRNA in IRF5⁻ and IRF5⁺ monocytes (*p=0.039, paired t-test).

B. Correlative analysis of GHITM and IRF5 mean expression from single-cell RNA-seq in 10-cell bins of increasing IRF5 expression (Pearson, R²=0.28, p<0.0001). Data from Hildreth *et al*²⁷.

Response Figure 21. The IRF5-GHITM axis is conserved in human monocytes and adipose tissue macrophages (legend continued)

C. Correlative analyses of MFI of IRF5, JC1-Green (mitochondrial mass, Mt Mass), JC1-Red (mΔΨ), and JC1-Red-to-Green ratio (mΔΨ/mass) in human visceral ATMs (vATM) from obese patients and in monocytes from patients with T2D (Mono) analysed by FACS (n=11 for monocytes, n=9 for ATMs, Pearson r).

D. University of California Santa Cruz (UCSC) genome browser (<http://genome.ucsc.edu>) tracks at the GHITM locus with tracks from JASPAR2020 to visualize transcription factor binding sites for IRF5 (session link).

E. Immunofluorescence staining of IRF5, oxidative phosphorylation (OXPHOS) enzyme complexes and CD14 in cytopsin prepared human monocytes from patients with T2D. Samples were separated based on IRF5 localization being either nuclear (Nuc) or cytoplasmic (Cyt). Quantification of OXPHOS staining in IRF5 Nuc and Cyt samples (n=10 per condition, unpaired t-test, *p=0.0335).

To complement these data and address function of GHITM in mitochondrial biogenesis and inflammation, we carried out an additional experiment in macrophages differentiated from the human Thp1 monocyte cell line. We applied siRNAs to target IRF5 (siIRF5) and/or GHITM (siGHITM) and analyzed mitochondrial respiration and gene expression. After confirming efficiency of siRNA-mediated knockdown (Response Figure 22A), we found siIRF5 increased maximal respiration in macrophages in response to palmitate, replicating our findings from mice (Response Figure 22B). Treatment with siGHITM or combination siIRF5 + siGHITM normalized respiration to the level of cells treated with a control siRNA (siCtrl). When analyzing gene expression, we found no effects on expression of NRF1 nor iNOS as makers affecting mitochondrial mass and function. Similarly, knockdowns had little-to-no effect on markers of anti-inflammatory polarization (IRF4, IL10). Inflammatory markers (TNF, IL6) were found to be induced by siGHITM. These findings confirm a role for GHITM in mitochondrial respiration in a model of human macrophages, and potentially an additional role in regulating inflammation.

Response Figure 22. GHITM knockdown alters mitochondrial respiration and expression of inflammatory genes in macrophages differentiated from human Thp1 cells. Thp1 cells were differentiated into macrophages, treated with siRNAs targeting IRF5 (siIRF5), GHITM (siGHITM) or a control siRNA (siCtrl) and subjected to palmitate treatment for 2 h.

A. IRF5 and GHITM gene expression quantified by qPCR.

B. Maximal respiration from extracellular flux analysis.

C. NRF1, iNOS, IRF4, IL10, TNF and IL6 expression quantified by qPCR.

Finally the mechanism whereby GHITM regulate inflammation might be out of the scope of this study but it seems like a low hanging fruit to measure ROS in the experiment where GHITM is knock-downed as the authors correctly suggest that ROS might be the link.

We thank the reviewer for this suggestion. Further investigating the mechanisms by which GHITM may interact with inflammatory functions is a very interesting area for future work. This area is partly addressed in the response to the prior comment (Response Figure 22); however, we do agree that further direct investigation of the mechanisms is out the scope of the current study. Here we characterized a mechanism by which IRF5, a pro-inflammatory transcription factor that has a role in insulin resistance^{6,19}, has a dual

role in limiting macrophage oxidative capacity in an early response to caloric excess. This limited oxidative capacity is associated with decreased expression of GHITM.

GHITM has been previously described to localize to the inner mitochondrial membrane and is important in maintaining mitochondrial architecture, mainly cristae, for efficient oxidative metabolism^{28,29}. Studies characterizing GHITM function have reported that siRNA-mediated knockdown results in loss of cristae organization and mitochondrial fragmentation^{28,29}. Indeed, such fragmentation is a well-described process that takes place in M1-like or pro-inflammatory macrophages. Interestingly, this has been described in response to both canonical M1-like polarizing agents (e.g. LPS) and in response to metabolic stimuli (e.g. palmitate)³⁰. Such mitochondrial fragmentation is associated with increased production of reactive oxygen species³¹. As for reports dedicated to GHITM regulation in inflammatory contexts, network-based analyses have associated dysregulated GHITM expression with infectious disease (HIV, SARS)³² and *in vitro* studies on Jurkat T cells have reported upregulation in response to certain chemokine signals³³.

To date relatively few studies have carried out in-depth investigations into the regulation and function of GHITM; 17 published works since its characterization in 2004. Here, we have added to the knowledgebase on this protein by describing a role for it in ATMs and BMDM. We have confirmed its functional importance in maintaining oxidative respiration and we also report potential for transcriptional regulation by IRF5. Further investigations centered around GHITM in inflammatory or metabolic disease contexts is a pertinent and interesting question, but is beyond the scope of the current work.

References

1. Lee, Y. S. *et al.* Inflammation is necessary for long-term but not short-term high-fat diet-induced insulin resistance. *Diabetes* **60**, 2474–2483 (2011).
2. Shimobayashi, M. *et al.* Insulin resistance causes inflammation in adipose tissue. *J. Clin. Invest.* **128**, 1538–1550 (2018).
3. Brunner, J. S. *et al.* The PI3K pathway preserves metabolic health through MARCO-dependent lipid uptake by adipose tissue macrophages. *Nat. Metab.* **2**, 1427–1442 (2020).
4. Jaitin, D. A. *et al.* Lipid-Associated Macrophages Control Metabolic Homeostasis in a Trem2-Dependent Manner. *Cell* **178**, 686–698.e14 (2019).
5. Coats, B. R. *et al.* Metabolically Activated Adipose Tissue Macrophages Perform Detrimental and Beneficial Functions during Diet-Induced Obesity. *Cell Rep.* **20**, 3149–3161 (2017).
6. Dalmas, E. *et al.* Irf5 deficiency in macrophages promotes beneficial adipose tissue expansion and insulin sensitivity during obesity. *Nat. Med.* **21**, 610–618 (2015).
7. Hedl, M., Yan, J. & Abraham, C. IRF5 and IRF5 Disease-Risk Variants Increase Glycolysis and Human M1 polarization By Regulating Proximal Signaling and Akt2 Activation. *Cell Rep.* **16**, 2442–2455 (2016).
8. Albers, G. J. *et al.* IRF5 regulates airway macrophage metabolic responses. *Clin. Exp. Immunol.* **204**, 134–143 (2021).
9. Virtue, S. & Vidal-Puig, A. GTTs and ITTs in mice: simple tests, complex answers. *Nat. Metab.* **3**, 883–886 (2021).
10. Gotoh, K. *et al.* Metabolic analysis of mouse bone-marrow-derived dendritic cells using an extracellular flux analyzer. *STAR Protoc.* **2**, 100401 (2021).
11. Seahorse XF Palmitate Oxidation Stress Test Kit FAO Substrate | Agilent. https://www.agilent.com/en/product/cell-analysis/real-time-cell-metabolic-analysis/xf-assay-kits-reagents-cell-assay-media/seahorse-xf-palmitate-oxidation-stress-test-kit-fao-substrate-911628#zoomELIBRARY_1001073.
12. Yoon, B. R., Oh, Y.-J., Kang, S. W., Lee, E. B. & Lee, W.-W. Role of SLC7A5 in Metabolic Reprogramming of Human Monocyte/Macrophage Immune Responses. *Front. Immunol.* **9**, (2018).
13. Noe, J. T. *et al.* Lactate supports a metabolic-epigenetic link in macrophage polarization. *Sci. Adv.* **7**, eabi8602.
14. Clausen, B. E., Burkhardt, C., Reith, W., Renkawitz, R. & Förster, I. Conditional gene targeting in macrophages and granulocytes using LysMcre mice. *Transgenic Res.* **8**, 265–277 (1999).
15. Li, T. *et al.* c-Rel Is a Myeloid Checkpoint for Cancer Immunotherapy. *Nat. Cancer* **1**, 507–517 (2020).

16. Deletion of the nuclear receptor ROR α in macrophages does not modify the development of obesity, insulin resistance and NASH | Scientific Reports. <https://www-nature-com.proxy.insermbiblio.inist.fr/articles/s41598-020-77858-6>.
17. Fan, R. *et al.* Loss of the co-repressor GPS2 sensitizes macrophage activation upon metabolic stress induced by obesity and type 2 diabetes. *Nat. Med.* **22**, 780–791 (2016).
18. Krausgruber, T. *et al.* IRF5 promotes inflammatory macrophage polarization and TH1-TH17 responses. *Nat. Immunol.* **12**, 231–238 (2011).
19. Weiss, M., Blazek, K., Byrne, A. J., Perocheau, D. P. & Udalova, I. A. IRF5 is a specific marker of inflammatory macrophages in vivo. *Mediators Inflamm.* **2013**, 245804 (2013).
20. Seneviratne, A. N. *et al.* Interferon Regulatory Factor 5 Controls Necrotic Core Formation in Atherosclerotic Lesions by Impairing Efferocytosis. *Circulation* **136**, 1140–1154 (2017).
21. IRF5 guides monocytes toward an inflammatory CD11c+ macrophage phenotype and promotes intestinal inflammation. <https://www-science-org.proxy.insermbiblio.inist.fr/doi/10.1126/sciimmunol.aax6085>.
22. Serbulea, V. *et al.* Macrophage phenotype and bioenergetics are controlled by oxidized phospholipids identified in lean and obese adipose tissue. *Proc. Natl. Acad. Sci. U. S. A.* **115**, E6254–E6263 (2018).
23. Hoofman, A. *et al.* The Immunomodulatory Metabolite Itaconate Modifies NLRP3 and Inhibits Inflammasome Activation. *Cell Metab.* **32**, 468–478.e7 (2020).
24. Mills, E. L. *et al.* Itaconate is an anti-inflammatory metabolite that activates Nrf2 via alkylation of KEAP1. *Nature* **556**, 113–117 (2018).
25. Mills, E. L. *et al.* Succinate Dehydrogenase Supports Metabolic Repurposing of Mitochondria to Drive Inflammatory Macrophages. *Cell* **167**, 457–470.e13 (2016).
26. Tannahill, G. M. *et al.* Succinate is an inflammatory signal that induces IL-1 β through HIF-1 α . *Nature* **496**, 238–242 (2013).
27. Hildreth, A. D. *et al.* Single-cell sequencing of human white adipose tissue identifies new cell states in health and obesity. *Nat. Immunol.* **22**, 639–653 (2021).
28. Oka, T. *et al.* Identification of a novel protein MICS1 that is involved in maintenance of mitochondrial morphology and apoptotic release of cytochrome c. *Mol. Biol. Cell* **19**, 2597–2608 (2008).
29. Reimers, K., Choi, C. Y., Bucan, V. & Vogt, P. M. The Growth-hormone inducible transmembrane protein (Ghitm) belongs to the Bax inhibitory protein-like family. *Int. J. Biol. Sci.* **3**, 471–476 (2007).
30. Zezina, E. *et al.* Mitochondrial fragmentation in human macrophages attenuates palmitate-induced inflammatory responses. *Biochim. Biophys. Acta Mol. Cell Biol. Lipids* **1863**, 433–446 (2018).
31. Alterations in mitochondrial morphology as a key driver of immunity and host defence. *EMBO Rep.* **22**, e53086 (2021).
32. Moni, M. A. & Liò, P. Network-based analysis of comorbidities risk during an infection: SARS and HIV case studies. *BMC Bioinformatics* **15**, 333 (2014).
33. Nagel, J. E. *et al.* Identification of genes differentially expressed in T cells following stimulation with the chemokines CXCL12 and CXCL10. *BMC Immunol.* **5**, 17 (2004).

REVIEWERS' COMMENTS

Reviewer #1 (Remarks to the Author):

The authors did a great effort to answer all reviewer comments and thereby strongly improved the quality of the manuscript. I don't have further questions.

Reviewer #2 (Remarks to the Author):

The authors have performed a tremendous amount of work for this revised version of the manuscript. The text has been clarified and reflects the suggestions of the reviewers. This reviewer is satisfied with the revised manuscript and has no further comment.